# Woody Litter Increases Headwater Stream Metal Export Ratio in an Alpine Forest

**Ziyi Liang, Fuzhong Wu, Xiangyin Ni, Bo Tan, Li Zhang, Zhenfeng Xu, Junyi Hu and Kai Yue \***

Long-Term Research Station of Alpine Forest Ecosystem, Provincial Key Laboratory of Ecological Forestry Engineering, Institute of Ecology and Forestry, Sichuan Agricultural University, Chengdu 611130, China; chn_liangzy@163.com (Z.L.); wufzchina@163.com (F.W.); nixiangyin_922@163.com (X.N.); bobotan1984@163.com (B.T.); zhangli16830116@hotmail.com (L.Z.); sicauxzf@163.com (Z.X.); hujunyi113@163.com (J.H.)
**\*** Correspondence: kkyue@fjnu.edu.cn; Tel.: +86-028-86291112

**Abstract:** Headwater streams have low productivity and are closely linked to forest ecosystems, which input a large amount of plant litter into streams. Most current studies have focused on the decomposition process of plant litter in streams, and the effects of non-woody and woody litter on metal transfer, accumulation, and storage in streams are poorly understood. Here, we addressed how non-woody and woody litter affect metals in headwater streams in an alpine forest on the Eastern Tibetan Plateau. This area is the source of many rivers and plays an important regulatory role in the regional climate and water conservation. Through comparisons of five metal concentrations, exports and storage in headwater streams with different input conditions of plant litter, our results showed that the input of woody litter could significantly increase flow discharge and increase the metal export ratio in the water. Similarly, the input of non-woody litter could reduce the metal concentration in the water and facilitate the stable storage of metals in the sediment in the headwater streams. Therefore, allochthonous non-woody and woody litter can affect the concentration of metals in water and sediment, and the transfer and accumulation of metals from upstream to downstream in headwater streams. This study provides basic data and new findings for understanding the effects of allochthonous plant litter on the accumulation and storage of metals in headwater forest streams and may provide new ideas for assessing and managing water quality in headwater streams in alpine forests.

**Keywords:** headwater stream; metals; non-woody litter; woody litter

## 1. Introduction

Metals are a critical and complex problem in ecosystems. Metals such as potassium (K), sodium (Na), calcium (Ca), and magnesium (Mg) are essential elements for living organisms, but they pose a threat to organisms when their concentrations are too high [1,2]. The concentrations of some trace metals in headwater streams require close monitoring, such as Fe, Mn, and Cr, because of their toxicity, persistence, tendency to bio-accumulate, and widespread presence in the environment [3]. Riparian vegetation, precipitation, and biogeochemical processes affect stream metal concentrations [4]. In the dry season, riparian plants can influence water quality by intercepting and absorbing nutrients and pollutants [5,6], thereby influencing the dynamics of element concentrations in sediments and streams [7]. Because headwater streams are relatively closely coupled to adjacent forests, they could receive large amounts of organic matter from riparian plants [8]. The forms of organic matter supplied by these riparian forests are divided into non-woody litter (fine litter, e.g., leaves, fruit, small bark

fragments, twigs, and flowers) and woody litter (coarse woody ≥ 10 cm and fine woody 1 cm ≤ diameter ≤ 10 cm, e.g., wood, branches, and roots) according to their ecological functions and diameters [9]. Previous studies have demonstrated that leaf and wood inputs are important material for forest stream food webs [10], and contribute to the retention of dissolved nutrients [11]. However, the effects of non-woody and woody litter inputs on the metals in streams are not clear, although studies have suggested that litter decomposition in streams is also an important metal cycling pathway [12].

Seventy-five percent of the global stream length is composed of headwater streams, and forest streams dominate watersheds worldwide [13,14]. Allochthonous plant litter not only represents an important terrestrial subsidy for these aquatic ecosystems, but also plays an essential role in supporting detritus-based food webs in headwater streams [15]. Once allothchonous plant litter is imported into a stream, it is either transported downstream by flow or retained on the streambed, where it is colonized by microorganisms and fragmented by invertebrates, broken down, decomposed, and converted into fine particulate organic matter [16]. Studies have shown that the decomposition rate of leaves in streams is faster than that on the forest floor [17], and broad, soft or senescence leaves can quickly release elements by enhancing microbial activity, thus significantly influencing the water quality in headwater streams [18]. Previous studies have shown that woody litter can affect the morphology and biological functioning of streams by creating new instream habitats [19], and reducing the energy of water during high discharge events [20]. These studies have shown that the input of non-woody and woody litter has an important influence on headwater streams and deserves further study. Moreover, woody litter can intercept and store organic matter and sediments [21], and most metals are transported in association with suspended sediment because of the affinity of metals for the common components of suspended particles [22,23]. Here, we address a hypothesis that there are differences in the contribution of woody litter input and non-woody litter input on the metal concentration and export in the headwater streams of alpine forests.

To test this hypothesis, we conducted a field control experiment by controlling the presence or absence of non-woody or woody litter inputs in headwater streams in an alpine coniferous forest. We compared the variations in the metal concentrations and exports in the headwater streams by removing the inputs of non-woody litter or woody litter in an alpine forest on the Eastern Tibetan Plateau. The objectives were: (1) To evaluate the effects of non-woody and woody litter on the metal concentrations and exports in the headwater streams of an alpine forest and (2) to determine whether non-woody and woody litter are key to controlling the metal concentrations in water and the storage in sediment.

## 2. Material and Methods

### 2.1. Study Site and Experimental Design

The study was conducted at the Long-term Research Station of Alpine Forest Ecosystems, Miyaluo Nature Reserve (102°53′–102°57′ E, 31°14″-31°19′ N, 2458–4169 m a.s.l. (above sea level)), which is located in Li County, Sichuan, southwestern China. This region is in a transitional zone between the Tibetan Plateau, the Sichuan Basin and the upper Yangtze River. The mean annual air temperature ranges from 2 °C to 4 °C, and the maximum and minimum temperatures are 23 °C and −18 °C, respectively. The mean annual precipitation is approximately 850 mm [24]. Meanwhile, the amount of water in the stream is greater in summer, and smaller or even dry in winter, which is characteristic of a typical seasonal forest stream. The study site is an alpine coniferous forest, and the dominant tree species are *Abies faxoniana* Rehd. and *Picea likiangensis* (Franch) Pritz var. *balfouriana* (Rehd·et Wils) Hillier ex Slavin, and associated species include *Cerasus duclouxii* (Koehne) Yu et Li, *Sabina saltuaria* (Rehd. et Wils.) Cheng et W.T. Wang, and *Betula albosinensis* Burk., interspersed with shrubs are composed of *Salix paraplesia* Schneid., *Rosa omeiensis* Rolfe, and *Rhododendron moupinense* Franch. [25].

To compare the effects of the inputs of non-woody litter and woody litter on streams, it was necessary to ensure that the characteristic conditions of the control streams were consistent and to

eliminate the effects of heterogeneity. A common persistent stream that is representative of a typical stream in the alpine coniferous forest was selected, and 6 straight streams with the similar characteristics were artificially excavated near this stream (3 replicates × (1 litter exclusion stream + 1 litter input stream), Figure 1). Water in the selected persistent natural stream was then introduced into the 6 streams. These 6 artificially excavated streams had the same length (50 m), width (0.5 m), and depth (0.15 m), and the streams were at a minimum 2 m apart. In addition, the slope, altitude (3600 m), substrate (mixing of clay, fine sand and gravel), and riparian vegetation of each artificial stream were similar to the original stream.

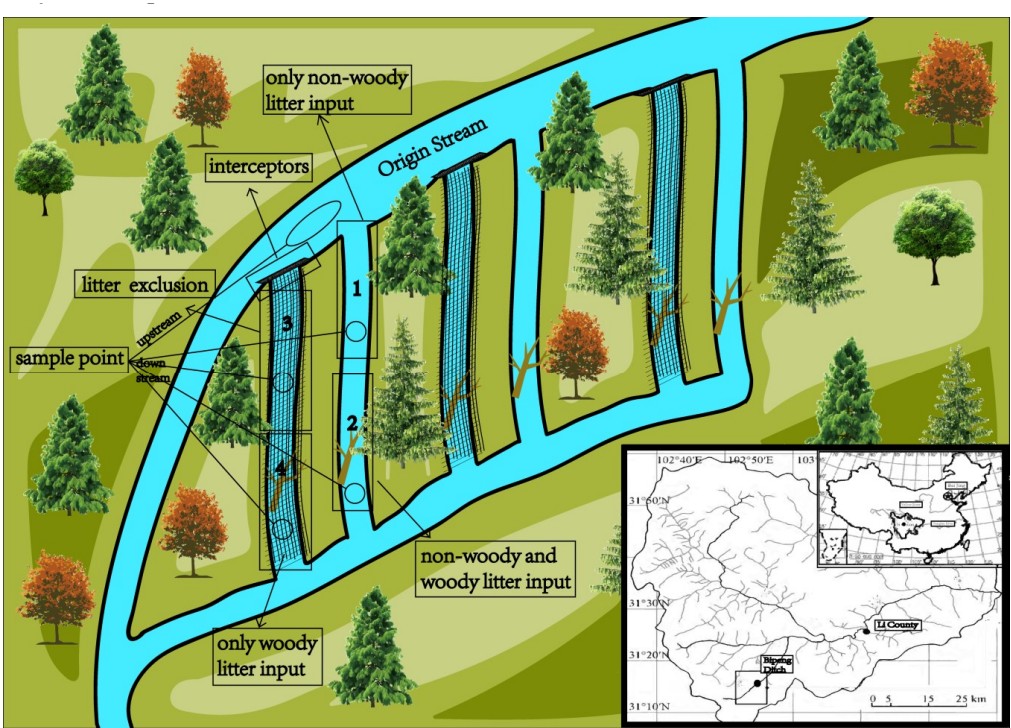

**Figure 1.** Schematic diagram showing the location and experimental design of our study in the alpine forest.

These artificially excavated streams were divided into two sections (25 m in each section). Woody litter was added artificially in the lower part of each stream, and this section of the stream was used as the woody litter input section. The artificially added woody litter consisted of large dead branches from nearby riparian vegetation. The woody litter was equally distributed into each stream to ensure the same control conditions.

Three of these artificial streams were selected as plant litter exclusion streams. A canopy of 1 mm mesh netting that did not affect light transmission was placed above these exclusion streams. This canopy covered the entirety of the exclusion streams and extended over both sides of the stream bank to exclude falling leaves and wood as well as lateral litter input. Meanwhile, interceptors with apertures of 1 mm and 5 mm were erectly installed at the inlet of the exclusion streams. These interceptors were installed to exclude plant litter inputs from the upper stream section, and two apertures were designed to prevent excessive litter that could affect the water flow characteristics from clogging the intercepting net.

After the above treatments, these streams could be divided into four treatments: (1) a stream with only non-woody litter input (the upper half of the reference stream); (2) a stream with non-woody and woody litter inputs (the lower half of the reference stream); (3) a stream with plant litter exclusion (the upper half of the exclusion stream); and (4) a stream with only woody litter input (the lower half of the exclusion stream). These 4 streams with different plant litter input conditions were regarded as

downstream streams, and the original streams leading out of these streams were regarded as upstream streams. By comparing the difference in the metal concentration and export between the upstream and downstream, we could determine the influence of the woody litter and non-woody litter inputs on the metals in the forested headwater streams.

### 2.2. Sample Collection and Statistical Analyses

According to previous research and phenological observations, the growing season in this region is from May to October [26]. After entering the snow period, restricted flow and less fallen litter will affect stream characteristics and plant litter inputs. Therefore, the sampling time was chosen from June to October 2017, once a month. Accumulated non-woody and woody litter were removed from the net before each sampling event. The water samples were collected randomly at approximately 1/2 depth of the stream using the pre-cleaned polyethylene bottles, and was carefully taken to avoid disturbing the bottom sediments and collecting surface floats. 2 L water samples were collected at each sample point, and then taken back to the laboratory within 24 hours and stored at 4 °C for chemical analysis. The surface sediments were collected using the pre-cleaned polyethylene bottles at each sample point by a five-point sampling method (use a random number table and tape measure to choose 5 points along the stream reach at random) [27]. After collection, the samples were returned to the laboratory and determined the concentrations of potassium (K), magnesium (Mg), iron (Fe), manganese (Mn), and chromium (Cr), digested using CEM-MARS 5, then tested using inductively coupled plasma spectroscopy (ICP-MS, IRIS Advantage 1000; Thermo Elemental, Waltham, MA, USA) [28].

At each sampling event, the metal exports per unit area of water were calculated as:

$$E = c \times F \tag{1}$$

where $E$ is the metals exports of water in the stream (mg·day$^{-1}$), and $c$ is the metals concentration in the water in the stream (µg·L$^{-1}$), $F$ is the flux of the stream (m$^3$·day$^{-1}$).

The metals storage per unit area of sediment was calculated as:

$$M = \frac{c \times m}{S} \tag{2}$$

where $M$ is the metals storage of sediment in the stream (g m$^{-2}$), $c$ is the metals concentration in the sediment in the stream (g kg$^{-1}$), $m$ is the amount of sediment in the stream (kg), and $S$ is the surface area of the sample at each stream (m$^2$).

The amount of sediment in the stream was calculated as:

$$m = \rho_s \times l \times w \times h_s \tag{3}$$

where $m$ is the amount of sediment in the stream (g), $\rho_s$ is the density of sediment (g m$^{-3}$), $l$ is the length of stream (m), $w$ is the width of stream (m), and $h_s$ is the depth of sediment in the stream (m).

The density of sediment was calculated as:

$$\rho_s = \frac{m_0}{(1 - m_c) \times V} \tag{4}$$

where $\rho_s$ is the density of sediment (g m$^{-3}$), $m_0$ is the drying weight of sediment (g), $m_c$ is the water content of sediment (%), and $V$ is the volume of sediment (m$^3$). The sediments used to calculate density were collected in a volume of 25 cm$^3$ polyethylene bottles at each point.

The cumulative exports rate of metals in the water was calculated as:

$$r = \frac{E_i - E_o}{E_o} \times 100\% \tag{5}$$

where *r* is the cumulative exports rate of metal in the water (%), $E_i$ is the metals exports from streams with different plant litter input conditions during the study period (g day$^{-1}$), $E_o$ is the metals exports from the origin of streams during the study period (g day$^{-1}$).

Analysis of variance (ANOVA) was used to test the effects of litter input conditions on water characteristics of the study streams. Repeated-measure ANOVA with Tukey's HSD was performed to test the effects of time and litter input conditions on metal concentrations and exports in the water, and metal concentrations and storages in the sediment. Spearman's correlation was selected for test the correlation coefficients between the environmental factors and metal concentrations and exports in the water, and metal concentrations and storages in the sediment [29,30]. All statistical analyses were performed using SPSS 22.0 (IBM SPSS Statistics Inc., Chicago, IL, USA). The water characteristics of the study streams were shown as the average (±SE, *n* = 15) during the study period (Table 1).

**Table 1.** Water characteristics of the study streams (average values during the study period, mean ± SE, *n* = 15).

| Streams | Temperature (°C) | Dissolved Oxygen (mg/L) | Conductivity (μs/cm) | pH | Illumination (lx) | Discharge (L/s) |
|---|---|---|---|---|---|---|
| origin | 6.87 ± 1.20 a | 7.60 ± 0.23 a | 23.62 ± 5.13 a | 6.42 ± 0.22 a | 12,256 ± 11,061 a | 6.09 ± 0.95 c |
| non-woody litter | 6.87 ± 1.20 a | 7.60 ± 0.23 a | 23.62 ± 5.13 a | 6.42 ± 0.22 a | 13,422 ± 14,124 a | 6.43 ± 1.46 c |
| non-woody and woody litter | 6.71 ± 0.65 a | 7.62 ± 0.27 a | 23.85 ± 4.07 a | 6.54 ± 0.21 a | 7981 ± 6294 b | 9.28 ± 1.79 a |
| litter exclusion | 6.65 ± 0.64 a | 7.51 ± 0.24 a | 24.87 ± 6.37 a | 6.44 ± 0.21 a | 12,405 ± 14,637 a | 6.91 ± 1.83 bc |
| woody litter | 6.71 ± 0.68 a | 7.57 ± 0.28 a | 24.69 ± 3.81 a | 6.40 ± 0.28 a | 8077 ± 5343 b | 8.53 ± 1.81 ab |

Different lowercase letters in the same column denote significant (*p* < 0.05) differences among different litter input conditions based on one-way ANOVA followed by multiple comparisons.

## 3. Results

### 3.1. Dynamics of Metal Concentrations in Water

The order of the concentrations of the metals in the water was K > Mg > Fe > Cr > Mn (Figure 2). Results of the repeated-measures ANOVA showed that time had significant impacts on all metals, while the different plant litter input conditions had significant impacts on K, Mn and Cr concentration (Table 2). While time had a significant effect on the concentrations of the metals in water, the influence pattern of time on the concentrations of these metals was different. Moreover, the concentrations of these metals in the water also had different responses to the input of different plant litter conditions. The exclusion of non-woody litter increased the K concentration in the water downstream. The Mn concentration was reduced in water downstream when woody litter was added to the streams together with non-woody litter. The Mg and Fe concentration of the water was relatively stable, and the allochthonous plant litter input had little effect on the concentration. However the Mg concentration in the water gradually increased during the study period, and the Fe concentration in the water decreased in September and October during the stream with exclusion of non-woody litter. The K concentration was positively correlated with the temperature and dissolved oxygen; the Mg concentration was negatively correlated with the dissolved oxygen and pH; and the Fe concentration was positively correlated with the temperature and conductivity (Table 3).

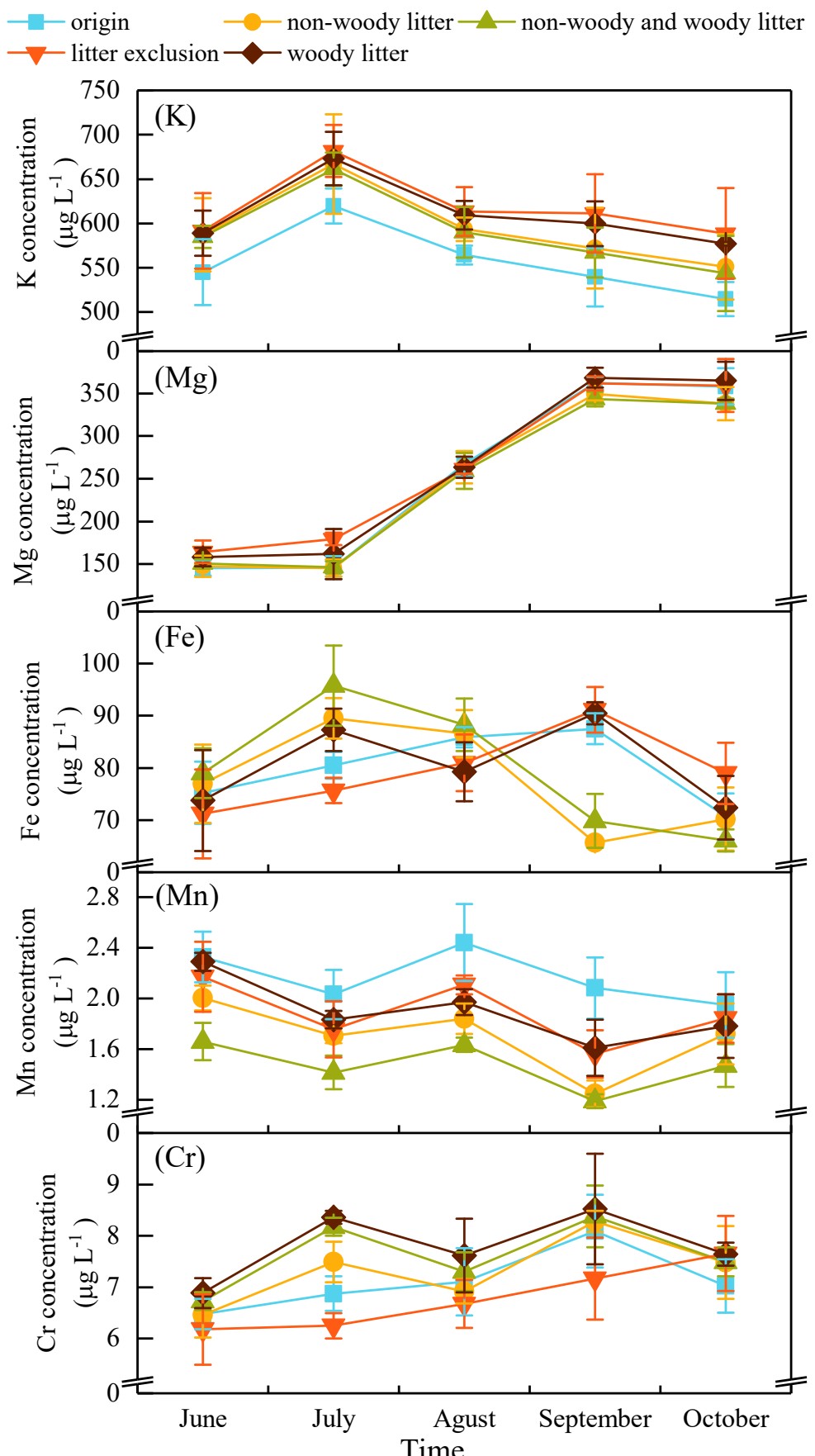

**Figure 2.** The K, Mg, Fe, Mn and Cr concentration of the water in the stream with different plant litter input conditions.

**Table 2.** Effects of time, debris input and their interaction on metals concentration of water tested by repeated-measure ANOVA analyses.

| Factor | df | K | | Mg | | Fe | | Mn | | Cr | |
|---|---|---|---|---|---|---|---|---|---|---|---|
| | | *F*-Value | *p*-Value | *F*-Value | *p*-Value | *F*-Value | *p*-Value | *F*-Value | *p*-Value | *F*-Value | *p*-Value |
| Time | 4 | 26.743 | <0.001 | 717.856 | <0.001 | 18.730 | <0.001 | 25.690 | <0.001 | 27.040 | <0.001 |
| Input | 4 | 5.042 | <0.05 | 3.186 | 0.062 | 0.740 | 0.586 | 21.203 | <0.001 | 3.778 | <0.05 |
| Time × Input | 16 | 0.194 | 1.000 | 0.829 | 0.647 | 6.582 | <0.001 | 1.194 | 0.314 | 2.268 | <0.05 |
| origin | | 556.67 b | | 255.84 a | | 79.96 a | | 2.17 a | | 7.11 ab | |
| non-woody litter | | 594.20 ab | | 248.72 a | | 77.77 a | | 1.70 bc | | 7.33 ab | |
| non-woody and woody litter | | 589.41 ab | | 247.57 a | | 79.80 a | | 1.47 c | | 7.62 ab | |
| litter exclusion | | 617.30 a | | 265.26 a | | 79.58 a | | 1.89 b | | 6.78 b | |
| woody litter | | 609.62 a | | 263.37 a | | 80.63 a | | 1.90 b | | 7.81 a | |

Different lowercase letters in the same column denote significant ($p < 0.05$) differences among different litter input conditions based on repeated-measure ANOVA followed by multiple comparisons.

**Table 3.** Correlation coefficients (*r*) between the environmental factors and the concentrations and exports of metals in the water.

| Factor | K | | Mg | | Fe | | Mn | | Cr | |
|---|---|---|---|---|---|---|---|---|---|---|
| | Concentration | Export | Concentration | Export | Concentration | Export | Concentration | Export | Concentration | Export |
| Temperature | 0.297 ** | 0.183 | −0.028 | 0.092 | 0.426 ** | 0.291 * | −0.054 | 0.041 | 0.136 | 0.159 |
| Dissolved Oxygen | 0.355 ** | 0.400 ** | −0.483 ** | −0.260 * | 0.054 | 0.281 * | −0.065 | 0.237 * | 0.034 | 0.258 * |
| Conductivity | 0.187 | 0.139 | −0.223 | −0.150 | 0.252 * | 0.208 | 0.188 | 0.226 | −0.199 | 0.016 |
| pH | −0.036 | −0.102 | −0.307 ** | −0.229 * | −0.113 | −0.128 | 0.078 | −0.021 | −0.112 | −0.079 |
| Illumination | −0.104 | −0.264 * | 0.174 | 0.011 | 0.144 | −0.147 | 0.173 | −0.135 | −0.150 | −0.257 * |

*, $p < 0.05$; **, $p < 0.01$, $n = 75$.

### 3.2. Dynamics of Metal Export in Water

The order of the metal export in the water was consistent with the concentration: K > Mg > Fe > Cr > Mn (Figure 3). Results of the repeated-measures ANOVA showed that time had significant impacts on Mg, Fe, Mn and Cr, while the input of non-woody and woody litter had no significant impacts on all metals export (Table 4). Nevertheless, we found that the metal export in the stream with added woody litter was greater than other streams (Figure 4), and the ratio of the export from the treated streams to that from origin stream indicated that woody litter can increase the K, Mg, Fe, Mn and Cr export in the water (Table 5). The K export was positively correlated with dissolved oxygen and negatively correlated with the illumination; the Mg export was negatively correlated with the dissolved oxygen and the pH; the Fe concentration was positively correlated with the temperature and the dissolved oxygen, and the Mn and Cr concentrations were positively correlated with the dissolved oxygen (Table 3).

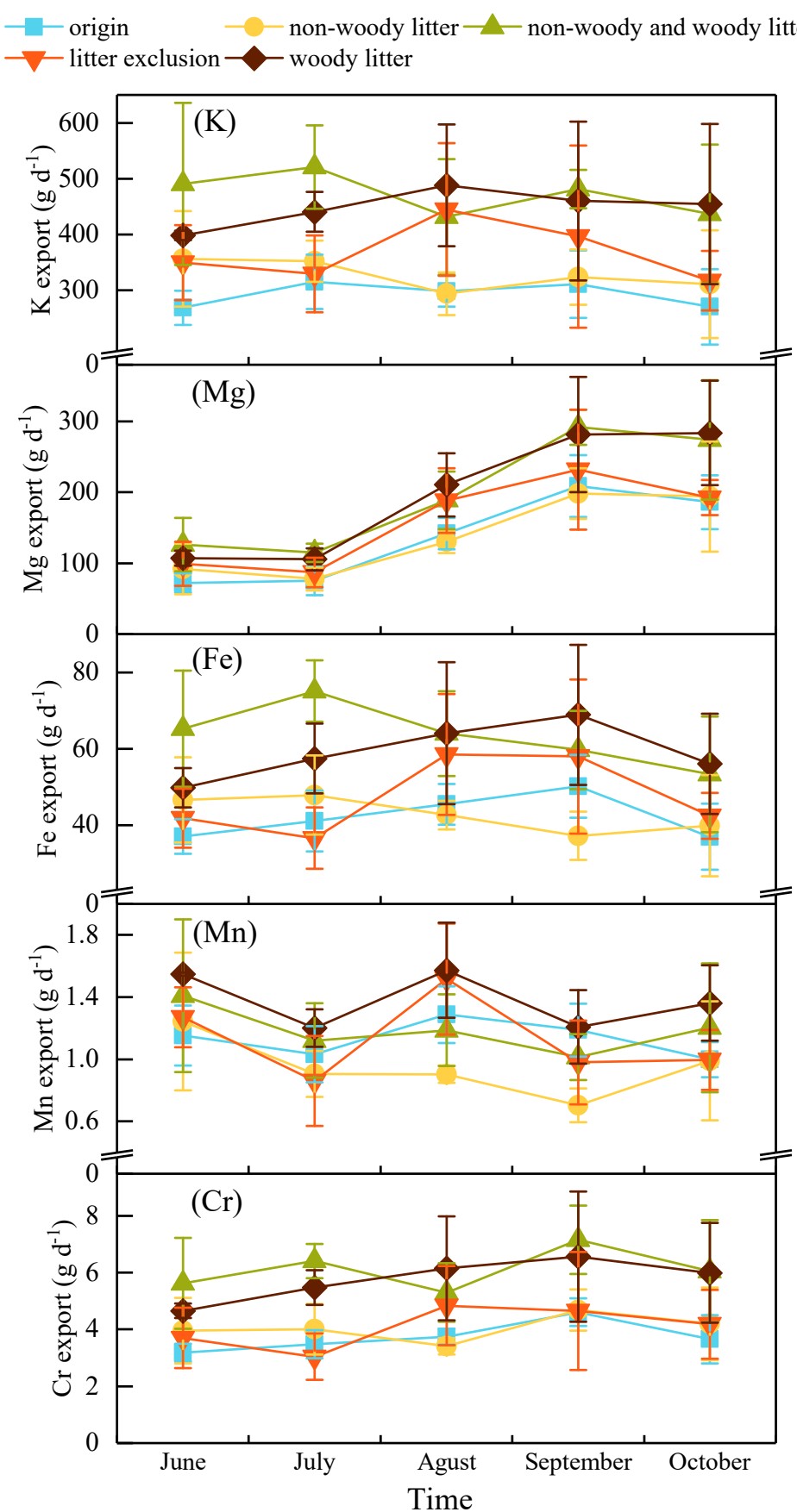

**Figure 3.** The K, Mg, Fe, Mn and Cr export of the water in the stream with different plant litter input conditions.

**Table 4.** Effects of time, debris input and their interaction on metals export of water tested by repeated-measure ANOVA analyses.

| Factor | df | K | | Mg | | Fe | | Mn | | Cr | |
|---|---|---|---|---|---|---|---|---|---|---|---|
| | | *F*-Value | *p*-Value | *F*-Value | *p*-Value | *F*-Value | *p*-Value | *F*-Value | *p*-Value | *F*-Value | *p*-Value |
| Time | 4 | 1.439 | 0.239 | 90.525 | <0.001 | 5.602 | <0.05 | 14.116 | <0.001 | 8.592 | <0.001 |
| Input | 4 | 3.098 | 0.067 | 2.070 | 0.160 | 3.024 | 0.071 | 1.646 | 0.238 | 3.284 | 0.058 |
| Time × Input | 16 | 1.516 | 0.142 | 1.229 | 0.324 | 3.189 | <0.05 | 2.077 | <0.05 | 1.520 | 0.140 |
| origin | | 292.28 a | | 136.82 a | | 42.13 a | | 1.13 a | | 3.74 a | |
| non-woody litter | | 327.04 a | | 138.41 a | | 42.81 a | | 0.95 a | | 4.05 a | |
| non-woody and woody litter | | 472.06 a | | 199.19 a | | 63.44 a | | 1.19 a | | 6.11 a | |
| litter exclusion | | 367.12 a | | 159.86 a | | 47.46 a | | 1.12 a | | 4.08 a | |
| woody litter | | 448.14 a | | 197.68 a | | 59.23 a | | 1.38 a | | 5.77 a | |

Different lowercase letters in the same column denote significant ($p < 0.05$) differences among different litter input conditions based on repeated-measure ANOVA followed by multiple comparisons.

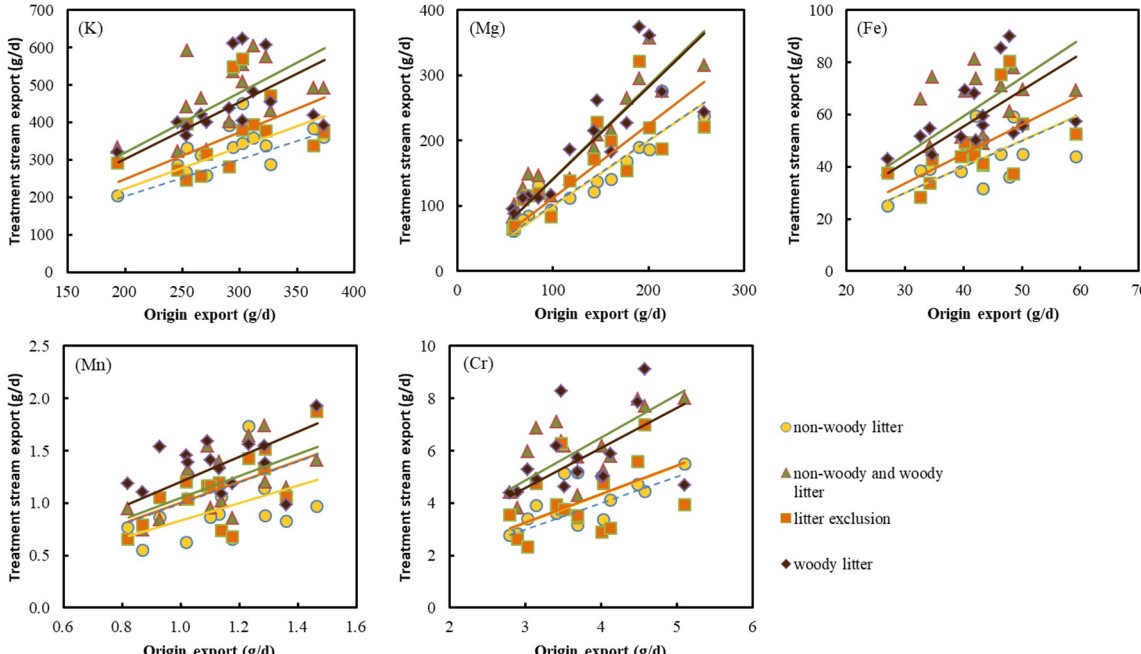

**Figure 4.** Export ratios of the metals (K, Mg, Fe, Mn, Cr) from the origin stream compared to those from the treatment stream (4 input conditions). The lines are the *k*-value trend lines of the treatment/origin streams, and the blue dotted line is the trend line with *k*-value of 1.

**Table 5.** Export ratios of the metals from the origin stream compared to the treatment stream with 4 litter input conditions.

| | K | Mg | Fe | Mn | Cr |
|---|---|---|---|---|---|
| origin | 1.00 c | 1.00 b | 1.00 b | 1.00 b | 1.00 b |
| non-woody litter | 1.12b c | 1.04 b | 1.03 b | 0.84 b | 1.09 b |
| non-woody and woody litter | 1.63 a | 1.50 a | 1.53a | 1.05 a | 1.65 a |
| litter exclusion | 1.27 b | 1.21 b | 1.13 b | 0.99 b | 1.10 b |
| woody litter | 1.55 a | 1.47 a | 1.43 a | 1.24 a | 1.56 a |

Different lowercase letters in the same column denote significant ($p < 0.05$) differences among different litter input conditions based on one-way ANOVA followed by multiple comparisons.

### 3.3. Dynamics of Metals Concentration in Sediment

The order of the concentrations of the metals in the sediment was K > Fe > Mg > Mn > Cr (Figure 5). Results of the repeated-measures ANOVA showed that time had significant impacts on all metals concentration of sediment, and different litter input conditions had significant impacts on Mg,

Fe, Mn and Cr (Table 6). Throughout the study period, the metals concentration of the sediment in the streams gradually decreased from June to October. The input of woody litter increased the Mg, Fe and Mn concentration of the sediment downstream. The variation of Cr concentration was obviously different from that of other metals. Compared with the origin stream, the exclusion of plant litter had no significant effect on the Cr concentration, and the stream with only non-woody litter had a greater concentration than others. The K and Fe concentration of the sediment was positively correlated with the conductivity; the Mg and Fe concentration of the sediment was positively correlated with the dissolved oxygen; and the Mn and Cr concentration of the sediment was positively correlated with the pH (Table 7).

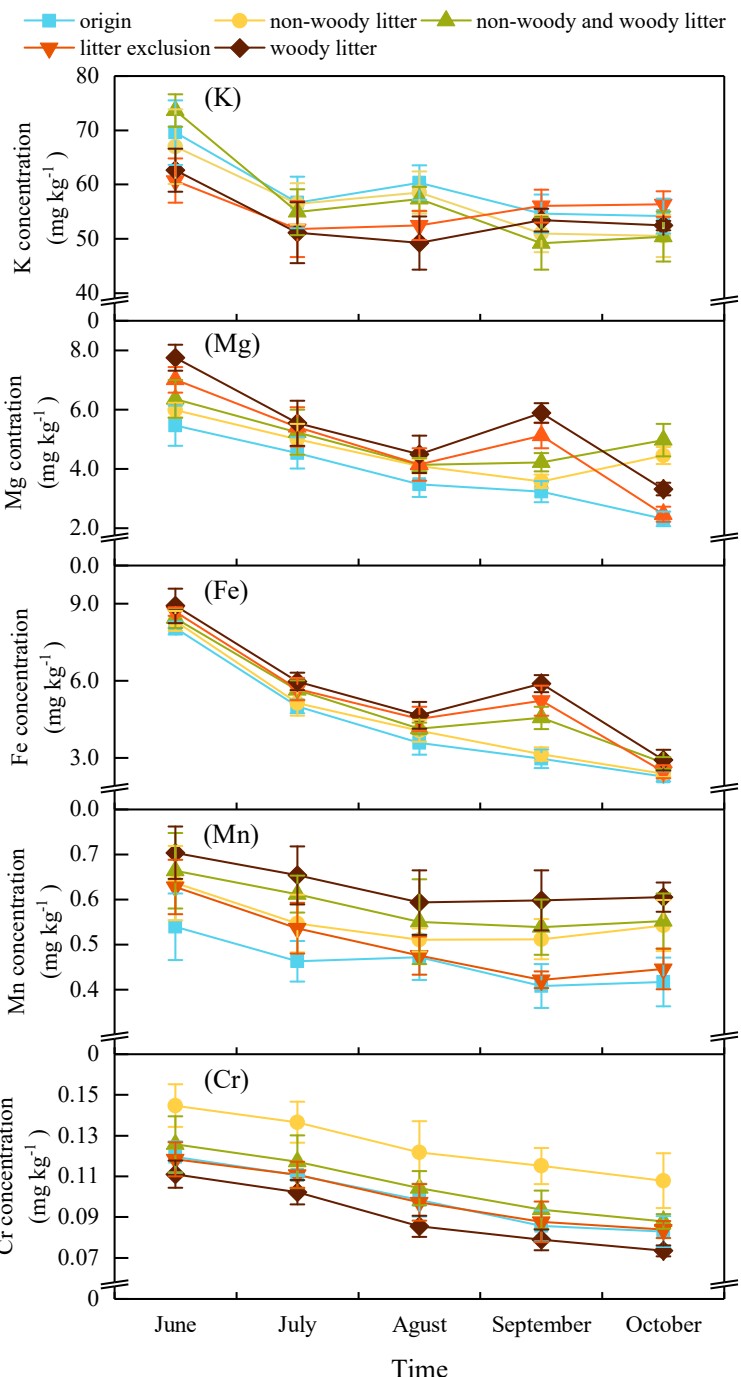

**Figure 5.** The K, Mg, Fe, Mn and Cr concentration of the sediment in the stream with different plant litter input conditions.

**Table 6.** Effects of time, debris input and their interaction on metals concentration of sediment tested by repeated-measure ANOVA analyses.

| Factor | df | K | | Mg | | Fe | | Mn | | Cr | |
|---|---|---|---|---|---|---|---|---|---|---|---|
| | | *F*-Value | *p*-Value | *F*-Value | *p*-Value | *F*-Value | *p*-Value | *F*-Value | *p*-Value | *F*-Value | *p*-Value |
| Time | 4 | 45.982 | <0.001 | 133.525 | <0.001 | 487.003 | <0.001 | 34.831 | <0.001 | 110.740 | <0.001 |
| Input | 4 | 1.541 | 0.264 | 10.342 | <0.001 | 27.993 | <0.001 | 5.577 | <0.05 | 9.888 | <0.01 |
| Time × Input | 16 | 3.334 | <0.001 | 9.657 | <0.001 | 3.968 | <0.001 | 0.993 | 0.483 | 0.118 | 1.000 |
| origin | | 59.08 a | | 3.80 b | | 4.37 c | | 0.46 b | | 0.10 b | |
| non-woody litter | | 56.71 a | | 4.62 ab | | 4.61 c | | 0.55 ab | | 0.13 a | |
| non-woody and woody litter | | 57.10 a | | 4.98 a | | 5.11 b | | 0.58 ab | | 0.11 b | |
| litter exclusion | | 55.49 a | | 4.83 a | | 5.31 ab | | 0.50 ab | | 0.10 b | |
| woody litter | | 53.78 a | | 5.40 a | | 5.67 a | | 0.63 a | | 0.90 b | |

Different lowercase letters in the same column denote significant ($p < 0.05$) differences among different litter input conditions based on repeated-measure ANOVA followed by multiple comparisons.

**Table 7.** Correlation coefficients (*r*) between the environmental factors and the concentrations and storages of metals in the sediment.

| Factor | K | | Mg | | Fe | | Mn | | Cr | |
|---|---|---|---|---|---|---|---|---|---|---|
| | Content | Storage | Content | Storage | Content | Storage | Content | Storage | Content | Storage |
| Temperature | −0.153 | 0.410 ** | −0.079 | 0.399 ** | 0.035 | 0.363 ** | −0.157 | 0.472 ** | −0.022 | 0.498 ** |
| Dissolved Oxygen | 0.207 | 0.161 | 0.233 ** | 0.255 * | 0.274 * | 0.289 * | −0.018 | 0.082 | 0.195 | 0.192 |
| Conductivity | 0.350 ** | 0.286 * | 0.089 | 0.235 * | 0.276 * | 0.354 ** | 0.095 | 0.226 | 0.197 | 0.273 * |
| pH | −0.025 | −0.256 * | 0.110 | −0.189 | 0.072 | −0.150 | 0.228 * | −0.186 | 0.275 * | −0.128 |
| Illumination | −0.049 | 0.398 ** | −0.192 | 0.284 * | −0.017 | 0.334 ** | −0.138 | 0.414 ** | 0.009 | 0.440 ** |

*, $p < 0.05$; **, $p < 0.01$, $n = 75$.

## 3.4. Dynamics of Metals Storage in Sediment

The order of the storage of the metals in the sediment was K > Fe > Mg > Mn > Cr (Figure 6). Results of the repeated-measures ANOVA showed that time had significant impacts on all metals storage, but different litter input conditions had no significant impacts on all metals storage (Table 8). The amount of metal storage in the sediment was not completely dependent on the metal concentration, as the element concentration in June were the highest during the study period, but their storage were not so. For K, Mn and Cr, the storage in July, August and September was greater than that in June and October. Regarding the time scale, the variation in the metal storage in the sediments after non-woody litter exclusion was greater than that in the streams without non-woody litter exclusion (Figure 6). Temperature and illumination were the dominant environmental factors affecting all metal storage in the sediment in the streams. The K, Mg, Fe, and Cr storage was positively correlated with conductivity; the Mg and Fe storage was positively correlated with dissolved oxygen; and the K storage in the sediment was negatively correlated with pH in the streams (Table 7).

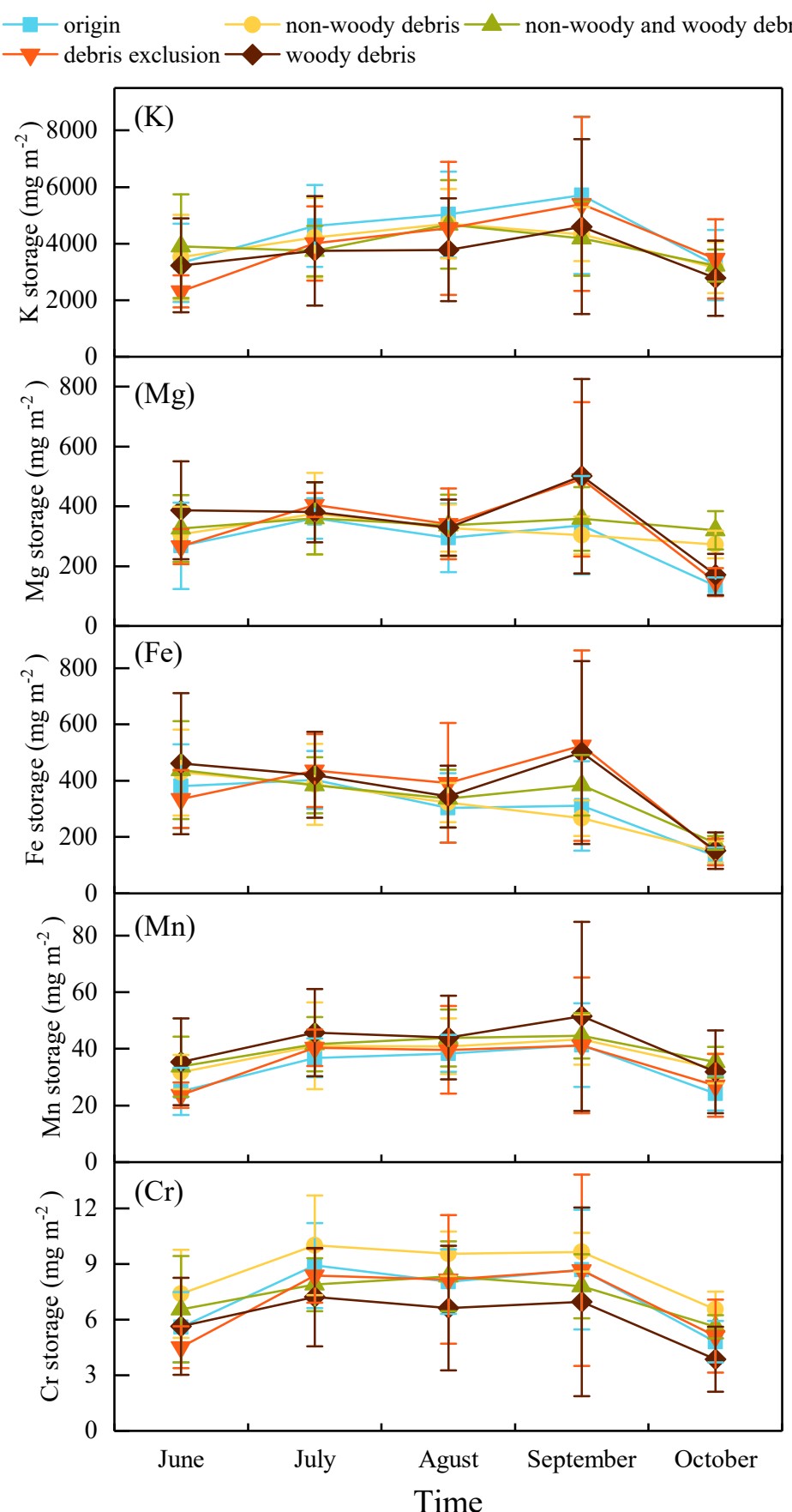

**Figure 6.** The K, Mg, Fe, Mn and Cr storage of the sediment in the stream with different plant litter input conditions.

**Table 8.** Effects of time, debris input and their interaction on metals storage of sediment tested by repeated-measure ANOVA analyses.

| Factor | df | K | | Mg | | Fe | | Mn | | Cr | |
|---|---|---|---|---|---|---|---|---|---|---|---|
| | | F-Value | p-Value | F-Value | p-Value | F-Value | p-Value | F-Value | p-Value | F-Value | p-Value |
| Time | 4 | 10.331 | <0.001 | 11.465 | <0.001 | 20.406 | <0.001 | 12.174 | <0.001 | 16.291 | <0.001 |
| Input | 4 | 0.099 | 0.980 | 0.236 | 0.912 | 0.184 | 0.942 | 0.309 | 0.866 | 0.561 | 0.697 |
| Time × Input | 16 | 0.785 | 0.692 | 1.503 | 0.219 | 1.046 | 0.434 | 0.235 | 0.999 | 0.400 | 0.975 |
| origin | | 4385.02 a | | 278.54 a | | 306.11 a | | 33.12 a | | 7.22 a | |
| non-woody litter | | 3981.85 a | | 316.76 a | | 310.69 a | | 38.00 a | | 8.63 a | |
| non-woody and woody litter | | 3945.64 a | | 339.82 a | | 344.26 a | | 39.80 a | | 7.24 a | |
| litter exclusion | | 3944.64 a | | 329.85 a | | 367.09 a | | 34.39 a | | 6.97 a | |
| woody litter | | 3624.27 a | | 353.65 a | | 375.75 a | | 41.70 a | | 6.06 a | |

Different lowercase letters in the same column denote significant ($p < 0.05$) differences among different litter input conditions based on repeated-measure ANOVA followed by multiple comparisons.

## 4. Discussion

### 4.1. Concentration and Export of Metals in Water

Through comparisons of the concentrations and exports of K, Mg, Fe, Mn, and Cr in headwater streams with different input conditions of plant litter in an alpine forest, we assessed the dynamics of these metals and the effects of non-woody litter and woody litter. In accordance with our hypothesis, our results showed that non-wood debris and wood debris have different contributions to metal in streams. Although there was no significant difference in metal export among streams with different litter input conditions, the input of woody litter significantly increased the metal export ratio in the water. Woody litter formed a major biological pathway for the transfer of elements from riparian vegetation to streams [31], and the woody litter input may affect the export of metals in water by affecting the behavioral factors of metals in headwater streams, such as the hydrology, flow characteristics and properties of the metals themselves [32]. The litter input can change the ionic strength, thus altering metal solubility and mobility [33]. Researchers have stressed the role of woody litter in retaining sediments and organic matter, increasing habitat diversity and providing refuge for aquatic organisms [34], as woody litter decompose more slowly in water than in terrestrial ecosystems. Thus, it is difficult to increase the concentration and export of metals in water by decomposition itself, and to alter metal mobility by changing the water characteristics [12]. The study results showed that the input of woody litter significantly increased the water discharge in the headwater stream (Table 1), and this may be the main reason for the increase in the metal export in the water, because woody litter changed the original flow characteristic of the streams and formed a step [35]. Moreover, our results also showed that there was a significant correlated relationship between the metal export of water and dissolved oxygen. Because of the step formed by the input of woody litter, the process of water flow descending was accompanied by intense water-air exchange, which increases the dissolved oxygen in the downstream, thus affecting ecosystem metabolism and respiration [36].

Many previous experiments on litter decomposition in streams have shown that leaves can decompose rapidly and release elements [18] because submerged leaves experience microbial colonization, are shredded by invertebrates, and are physically abraded [37,38]; thus, non-woody litter may increase the concentration of metals in headwater streams. However, our results showed that the input of non-woody litter reduced the metal concentration in the water, and the streams with plant litter exclusion had higher metal export ratio than the streams with only non-woody litter, which indicates that non-woody litter can reduce the metal export, absorb metals from water and contribute to the self-purification capacity of headwater streams [12]. Direct fresh non-woody litter that falls typically has a lower density than water and moves downstream on the water surface until obstacles protruding above the water surface trap it [39,40]. This litter then becomes waterlogged, sinks, and accumulates on the stream bottom [41,42], and non-woody litter may intercept and absorb the metals from surface runoff during floating. While the input of non-woody litter reduced the metals concentration in the

water, the contribution of woody litter input was greater than that of non-woody litter input on the metal export.

*4.2. Concentration and Storage of Metals in Sediment*

By comparing the K, Mg, Fe, Mn, and Cr concentrations and storage of sediment in the headwater streams with different plant litter input conditions, we assessed the effects of non-woody litter and woody litter input for these metals in sediments. The results showed that although non-woody litter and woody litter could affect the metal concentration in the surface sediment, they had no significant effect on the storage of metals in the surface sediment. Compared with the streams with non-woody litter input exclusion, the variation in the metals in the sediment of the streams with non-woody litter input was more stable. Sediment dynamics include the entrainment, transport, deposition, and storage of particulate matter, which includes mineral sediment and particulate organic matter in this study [43]. Allochthonous non-woody and woody litter could influence the sediment dynamics in headwater streams. After non-woody litter is supplied to headwater streams by forest ecosystems, it is initially exposed to the water surface, passively adsorbing and actively taking up suspended sediments in the water [16]. Then, a substantial portion of the non-woody litter entering running waters is buried in the streambed after becoming submerged [44], because of relatively high sediment yields and flow velocities [45]. The input of non-woody litter can reduce the mobility and erodibility of sediments in headwater streams [46], and thus, the variation in the metal storage in the sediment downstream is more stable and is not significantly different from that in the sediment upstream. The exclusion of non-woody litter could cause sediment to move downstream, and thus, non-woody litter could have difficultly contributing to the metal storage in the sediment in headwater streams. The input of woody litter increased the Mg, Fe and Mn concentration and storage in the sediment. As a harmful heavy metal, Cr showed an immobilization pattern in the early stage of litter decomposition [47], and therefore, the concentration and storage of Cr in the sediment increased in the streams with non-woody litter input.

The results showed that the storage of all the metals in the sediment was significantly correlated with temperature and illumination ($p < 0.05$). The addition of woody litter could reduce the water temperature and illumination in the headwater stream. Plant litter input drives ecosystem functioning by promoting periods of intense respiratory activity [48]. High-quality plant litter is preferred by aquatic fungi, bacteria and plankton, but it can be affected when light and temperature are limited [16]. Illumination is a direct mediating factor on ecosystem metabolism [36], and temperature can influence organisms such as bacteria, fungi and microalgae in streams to respond quickly to environmental changes, thus driving a large portion of material cycles [49]. These factors play an important role in regulating primary productivity, decomposition, and disturbance. These factors are also influenced by the input of plant litter, and the exclusion of plant litter could reduce the water temperature in this study, thus affecting the transport, accumulation, and storage of metals in the sediment in the headwater streams.

## 5. Conclusions

Assessing the effects of non-woody litter and woody litter inputs on metals through comparisons of the concentrations, exports and storage of the metals in headwater streams with different plant litter input conditions in an alpine forest is critical to understanding the process controlling the metals in water and sediments by allochthonous plant litter. Through this method of control and comparison, it was found that the input of woody litter can significantly increase the metal export ratio in water. Meanwhile, the input of non-woody litter can reduce the metal concentration in water and facilitate the stable storage of metals in the sediment in the headwater streams. Although the input of woody and non-woody litter had different effects on metal concentration in sediments, they had no significant effect on metal storage in sediments. The dynamics of only five metals in the headwater streams with different input conditions of plant litter were measured in this study, but these metals are representative

and contain macroelement, microelement and heavy metal. The input of non-woody and woody litter had contrary effects on the storage of heavy metal Cr in the sediment from other elements. Nevertheless, the study results still showed the responses of the metals to the input of allochthonous organic litter. Knowing this response is critical for understanding the processes controlling the metal concentrations in water and sediments and for assessing and managing the water quality of headwater streams in the alpine forest.

**Author Contributions:** K.Y. and F.W. conceived the idea. Z.L. and J.H. designed the experiment. X.N., Z.X., B.T. and L.Z. provided advice in study design. Z.L. and J.H. conducted the fieldwork. Z.L. collected and analyzed the data. Z.L. wrote the manuscript. All authors contributed to the revision of the manuscript.

**Funding:** This study was financially supported by the National Natural Science Foundation of China (31800373 and 31670526), the National Key Technologies R & D Program of China (2017YFC0505003), the Key Technologies R & D Program of Sichuan (18ZDYF0307) and the Fok Ying-Tong Education Foundation for Young Teachers (161101).

**Acknowledgments:** We would like to thank Junwei Wu, Fan Yang, Zhuang Wang, Liyan Zhuang, Jiao Zhou and Fei Duan for their kind assistance in field work.

**Conflicts of Interest:** The authors declare no conflict of interest.

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
