# Peer review of "Woody Litter Increases Headwater Stream Metal Export Ratio in an Alpine Forest"

_forests, doi:10.3390/f10050379_

Reviewer 1 Report

The Authors carried out an interesting project, unfortunately have not devoted enough

effort to present the results in a clear and fully exhaustive way. Working on improving the

manuscript they may wish to take into account the following points:

1. The material of plant origin has been divided into woody and non-woody debris. However the

categories have not been precisely defined. Hence problematic inclusion of twigs and bark

into the “non-woody” category. [l.43]

2. The phrase “metal elements” is too windy, “metals” suffice.

3. A “long-term stable stream” should be precisely defined. [l.90]

4. The description of the experimental setup is insufficient. It is difficult to visualise the layout

of the artificial streams, their connection to the “origin” stream, and the sampling points. I

would suggest adding a figure with a sketch depicting the entire setup.

5. Sampling:

    5.1. How many water samples were taken at each sampling point?

    5.2. How the water samples were treated immediately after being collected?

    5.3. The five-point sampling method has not been sufficiently described. Therefore it needs

a reference to the original paper where it was presented. Adding an explanation how the

five-point sampling method was adapted to present research would help.

    5.4. How the “amount of sediment in the stream” was measured? [l.138]

6. The metal content of sediments is ambiguously defined. In the case of sediments L ≠ kg.

[l.138]

7. Data in Table 1 should be given with standard deviations.

8. The concept of “metal storage” in sediments is poorly defined (see also remark 5.4 above). The Authors should rethink it anew.

8.1. What is the difference between “storage” [l.198] and “content” [l.199]?

9. Figures:

    9.1. All the graphs are too small. The Authors should take into consideration that in print the colours may be gone.

    9.2. Asterisks in Fig. 3 are not explained. Presumably they mark significant differences, but

it is not clear to which differences they relate: between treatments or between months?

    9.3. Does the lack of asterisks in Fig. 1 mean that there were no significant differences

there?

10. The paper is in need of language revision because some fragments are incomprehensible.

Some examples have been marked directly in the text.

11. Minor issues have been marked directly on the manuscript.

Author Response

Dear editor

 We are very grateful for your comments and useful suggestions on our manuscript entitled "Woody litter increases headwater stream metal export in an alpine forest", (manuscript number: forests-470383). Those comments are valuable and very helpful for revising and improving our manuscript, as well as the important guiding significance to our research. Based on these comments and suggestions, we have revised the manuscript accordingly. The language has been further polished by American Journal Experts, which is an English language editing company (https://www.aje.com). All comments and suggestions provided by the reviewers are shown in black, and our responses are in blue. Revisions are marked in red in the main text. Our responses to each and all the comments are listed was follows:

 ü  The material of plant origin has been divided into woody and non-woody debris. However the categories have not been precisely defined. Hence problematic inclusion of twigs and bark into the “non-woody” category. [l.43]

Response: We thank you for the useful comments! Our classification criterion for woody litter is diameter > 2 mm, while twigs and barks are not within this range, so both are classified as non-woody litter. This have been clarified in the main text.

ü  The phrase “metal elements” is too windy, “metals” suffice.

Response: Corrected as suggested.

ü  A “long-term stable stream” should be precisely defined. [l.90]

Response: We have Modified “long-term stable stream” to “persistent stream”.

ü  The description of the experimental setup is insufficient. It is difficult to visualise the layout of the artificial streams, their connection to the “origin” stream, and the sampling points. I would suggest adding a figure with a sketch depicting the entire setup.

Response: Corrected as suggested, the schematic diagram of experimental design has been added, please see figure 1.

ü  How many water samples were taken at each sampling point?

Response: Corrected as suggested, relevant content has been added in the 2.2 sample collection, and 2L of water samples were collected at each sample point.

ü  How the water samples were treated immediately after being collected?

Response: Corrected as suggested, relevant content has been added in the 2.2 sample collection, and see the revised manuscript.

ü  The five-point sampling method has not been sufficiently described. Therefore it needs a reference to the original paper where it was presented. Adding an explanation how the five-point sampling method was adapted to present research would help.

Response: Corrected as suggested.

ü  How the “amount of sediment in the stream” was measured? [l.138]

Response: Relevant measurements methods and calculation formulas have been added as “m=ρs×l×w×hs” and “ρs=” in the 2.2.

ü  The metal content of sediments is ambiguously defined. In the case of sediments L ≠ kg. [l.138]

Response: The unit g L-1 has been deleted. The concentration here is the calculated value, not the directly measured concentration, so its unit should be g kg-1.

ü  Data in Table 1 should be given with standard deviations.

Response: Corrected as suggested.

ü  The concept of “metal storage” in sediments is poorly defined (see also remark 5.4 above). The Authors should rethink it anew.

Response: Relevant measurements methods and calculation formulas have been added as “M = ”, “m=ρs×l×w×hs” and “ρs=” in the 2.2, and see the revised manuscript.

ü  What is the difference between “storage” [l.198] and “content” [l.199]?

Response: The unit of “storage” is g m-2, and the unit of “concentration” is g kg-1, see the formula (2). We consider that the storage can better reflect the impact of an element. For example, in a watershed, the metal content in sediments is very high, but the sediment storage is very small, which is not enough to have an influence on the region.

ü  All the graphs are too small. The Authors should take into consideration that in print the colours may be gone.

Response: We have overstriking the lines in all figures, and each figure is the vector graph.

ü  Asterisks in Fig. 3 are not explained. Presumably they mark significant differences, but it is not clear to which differences they relate: between treatments or between months?

Response: We deleted the asterisk and added the results of repeated measurement variance analysis in the result section. Details please see the revised manuscript.

ü  Does the lack of asterisks in Fig. 1 mean that there were no significant differences there?

Response: We added the results of repeated measurement variance analysis to the result section.

ü  The paper is in need of language revision because some fragments are incomprehensible. Some examples have been marked directly in the text.

Response: The language has been further polished by American Journal Experts, which is an English language editing company (https://www.aje.com).

ü  Minor issues have been marked directly on the manuscript.

Response: Corresponding issues have been revised, and see the revised manuscript.

Reviewer 2 Report

The manuscript of Liang et al. examines the effects of woody and non-woody debris on metal export and storage in headwater streams. This study is novel and adds to our understanding of metal transport and storage in headwater streams. However, I have some major concerns that need to be addressed before publication, including justification of study design, clarification of experimental stream design, and inclusion of results of statistical tests.

The introduction provides an adequate summary of background information, and justifies the study. I only briefly read the discussion. The general conclusions and explanations seem appropriate, but more information is needed on the methods and results (see major concerns below) in order to assess the discussion more thoroughly. Once these are addressed, I would be happy to read it again and provide more details on the introduction and discussion sections.

Major Concerns

Experimental design and artificial streams

In L 88 you say control “streams” e.g. plural, but then in L90-91 mention that a single control stream was selected. Is a single control stream enough to compare to? Or are you only using this stream to match the morphological characteristics for your experimental streams (“origin” in Table 1)? Please clarify.

More detail is needed describing the artificial streams that were excavated. Were these excavated in the forest, and was the riparian vegetation and canopy cover similar to each other and / or to representative streams in this area? How many streams exactly were excavated? Do these streams meander or are they generally straight? Where does the water from these streams come from? Diversions from naturally occurring streams?

The description in L 91-92 is confusing. It seems like a total of 6 streams were excavated, but it’s unclear what the reference and exclusion stream refers to. Additionally, in the paragraph starting on L 95 it says that woody debris was added in the lower section. Is this for all streams? Does placing wood only in lower half affect your results? For example, would placing woody debris in upper section instead of lower section alter interaction between woody and non-woody debris? Due to the downstream flow characteristic of streams, I could foresee treatments in the upstream section affecting your observations in the downstream section. Comments or justification for this design are warranted. A figure of the stream design is desirable, and may clear up some of my confusion.

The wording of the different treatments throughout is hard to follow. I would recommend changing “non-woody” to something like “litter”. I realize that “non-woody” is inclusive and a more accurate description, but I believe it would make it easier on the reader.

In L 103-104 you mention apertures. I understand that these are to capture plant-litter from the exclusion streams, but do you think this could impact your results? You mention that you remove debris from apertures before sampling (L122), but I could envision this material leaching or absorbing metals from the water column in between sampling events. Please justify this sampling regime.

L 126 please explain the “five-point” sampling method, or provide a reference if this is a common and accepted method.

“Origin” stream. More information is needed on the origin stream (see comment on Figure 2 below). Were samples for metal concentration taken at the same time as experimental streams? What is the status of woody and non-woody debris in the origin stream? Is this essentially comparable to the “woody + non woody debris” experimental stream? Again, information on where the water for the experimental streams comes from would be helpful. For example, if it is diverted from origin stream, are the results in figure 2 directly observing how the treatments affect water from this stream?

Stats

I find the results section to be generally insufficient and lacking details and specifics of statistical testing.

In the paragraph beginning on L 145 you state that ANOVA and Tukey HSD tests were conducted, but I don’t believe these results are presented. Indeed, in L 241-242 you state that “woody debris significantly increased water discharge” and cite table 1, but Table 1 does not include any statistical results. Likewise, I see no mention of the Tukey results anywhere.

Table 1

Please add a description of variation to your averages e.g. standard deviation. I would also recommend adding a row with results of an ANOVA (or similar) test for each variable in the columns to show the reader if streams varied significantly (e.g. see Stats comment above) or were similar to one another. Also please clarify what the “origin” stream refers to.

Tables 2 and 3

Are these results for all streams, regardless of treatment type? Do these independent variables vary significantly between treatment types? i.e. see comment for Table 1 above.

Figure 1

Include stats on this figure. Or are there no significant differences

Figure 2

So samples of metal concentrations were also taken from “origin” stream? Were these taken at the same time and following same methods? Or was this compared to data previously collected? This needs to be explained in the methods section. I would also recommend adding a 1:1 line to each panel for reference. This will allow the reader to quickly assess which treatments increase or decrease export.

Figure 3

Explain what the star on the panels refers to. Statistical significance? What test?

Author Response

Dear editor

 We are very grateful for your comments and useful suggestions on our manuscript entitled "Woody litter increases headwater stream metal export in an alpine forest", (manuscript number: forests-470383). Those comments are valuable and very helpful for revising and improving our manuscript, as well as the important guiding significance to our research. Based on these comments and suggestions, we have revised the manuscript accordingly. The language has been further polished by American Journal Experts, which is an English language editing company (https://www.aje.com). All comments and suggestions provided by the reviewers are shown in black, and our responses are in blue. Revisions are marked in red in the main text. Our responses to each and all the comments are listed was follows:

 ü  In L 88 you say control “streams” e.g. plural, but then in L90-91 mention that a single control stream was selected. Is a single control stream enough to compare to? Or are you only using this stream to match the morphological characteristics for your experimental streams (“origin” in Table 1)? Please clarify.

Response: We thank you for the insightful comments and useful suggestions! In order to show our experimental design more clearly, we have added a schematic diagram of the experimental design in the revised manuscript, and see the Figure 1. We artificially excavated 6 streams with similar characteristics, and introduced water into the 6 streams form a natural permanent stream (the stream was selected). Each of 2 streams is a group, one of which is not treated as the reference, and the other stream is covered by nylon mesh to exclusion the non-woody input.

ü  More detail is needed describing the artificial streams that were excavated. Were these excavated in the forest, and was the riparian vegetation and canopy cover similar to each other and / or to representative streams in this area? How many streams exactly were excavated? Do these streams meander or are they generally straight? Where does the water from these streams come from? Diversions from naturally occurring streams?

Response: We have modified the experimental design to make it more detailed and clear. We selected a representative stream in an alpine forest, and artificially excavated 6 streams on the riparian of the stream. These 6 streams were generally straight, and the water was introduced from the selected stream.

ü  The description in L 91-92 is confusing. It seems like a total of 6 streams were excavated, but it’s unclear what the reference and exclusion stream refers to. Additionally, in the paragraph starting on L 95 it says that woody debris was added in the lower section. Is this for all streams? Does placing wood only in lower half affect your results? For example, would placing woody debris in upper section instead of lower section alter interaction between woody and non-woody debris? Due to the downstream flow characteristic of streams, I could foresee treatments in the upstream section affecting your observations in the downstream section. Comments or justification for this design are warranted. A figure of the stream design is desirable, and may clear up some of my confusion.

Response: We have added a schematic diagram of the experimental design in the revised manuscript, and see the Figure 1. We have modified the description, and now described as “litter exclusion stream” and “litter input stream”. We added woody litter in the lower part of all streams, which was an independent treatment. Considered that the addition of woody litter might change the characteristics of streams, it was necessary to add woody litter in the lower stream in order not to affect the separate effect of non-woody litter or the litter exclusion.

ü  The wording of the different treatments throughout is hard to follow. I would recommend changing “non-woody” to something like “litter”. I realize that “non-woody” is inclusive and a more accurate description, but I believe it would make it easier on the reader.

Response: The “non-woody” and “woody” here were intended to distinguish the influence of their input on the stream by the size. According to your suggestion, we decided to modify “non-woody debris” to “non-woody litter” and “woody debris” to “woody litter”. Woody litter generally refers to diameters > 2 mm, such as wood, branches and roots, and non-woody litter generally refers to diameters < 2 mm, such as leaves, fruit, bark, twigs and flowers. This have also been clarified in the revised manuscript.

ü  In L 103-104 you mention apertures. I understand that these are to capture plant-litter from the exclusion streams, but do you think this could impact your results? You mention that you remove debris from apertures before sampling (L122), but I could envision this material leaching or absorbing metals from the water column in between sampling events. Please justify this sampling regime.

Response: We also consider that the interceptor at the inlet of streams may affect the characteristics of the stream or others. Therefore, two kinds of intercepting nets with different apertures were set to reduce the influence of intercepting litter from the upstream.

ü  L 126 please explain the “five-point” sampling method, or provide a reference if this is a common and accepted method.

Response: Corrected as suggested, and see the revised manuscript.

ü  “Origin” stream. More information is needed on the origin stream (see comment on Figure 2 below). Were samples for metal concentration taken at the same time as experimental streams? What is the status of woody and non-woody debris in the origin stream? Is this essentially comparable to the “woody + non woody debris” experimental stream? Again, information on where the water for the experimental streams comes from would be helpful. For example, if it is diverted from origin stream, are the results in figure 2 directly observing how the treatments affect water from this stream?

Response: The “origin” is the natural stream before it flows into the artificially excavated stream, and is used to compare with the treated streams to survey what happens after the treatment. We added a blue dotted line in the figure 3(formerly figure 2), indicating a trend line with k-value of 1.

ü  I find the results section to be generally insufficient and lacking details and specifics of statistical testing.

Response: We have enriched the results section and added Table 2, 3, 5 and 6 for repeated-measure ANOVA analyses, and see the revised manuscript.

ü  In the paragraph beginning on L 145 you state that ANOVA and Tukey HSD tests were conducted, but I don’t believe these results are presented. Indeed, in L 241-242 you state that “woody debris significantly increased water discharge” and cite table 1, but Table 1 does not include any statistical results. Likewise, I see no mention of the Tukey results anywhere.

Response: We have enriched the results section and added Table 2, 3, 5 and 6 for repeated-measure ANOVA analyses, and added standard deviations to Table 1, see the revised manuscript.

ü  Table 1. Please add a description of variation to your averages e.g. standard deviation. I would also recommend adding a row with results of an ANOVA (or similar) test for each variable in the columns to show the reader if streams varied significantly (e.g. see Stats comment above) or were similar to one another. Also please clarify what the “origin” stream refers to.

Response: Corrected as suggested.

ü  Tables 2 and 3. Are these results for all streams, regardless of treatment type? Do these independent variables vary significantly between treatment types? i.e. see comment for Table 1 above.

Response: We have added standard deviation and ANOVA analyses in Table 1, because the different litter input conditions have no significant effect on these environmental factors, the results here did not distinguish between different treatments.

ü  Figure 1. Include stats on this figure. Or are there no significant differences

Response: We have added the repeated-measure ANOVA analyses for Figure 1, see Table 2 and 3.

ü  Figure 2. So samples of metal concentrations were also taken from “origin” stream? Were these taken at the same time and following same methods? Or was this compared to data previously collected? This needs to be explained in the methods section. I would also recommend adding a 1:1 line to each panel for reference. This will allow the reader to quickly assess which treatments increase or decrease export.

Response: The time and methods of sampling from “origin stream” are similar with other streams. Accorded as suggestion, we have added a blue dotted line of 1:1 to the figure for reference.

ü  Figure 3. Explain what the star on the panels refers to. Statistical significance? What test?

Response: We have deleted the asterisk, and added the repeated-measure ANOVA analyses for Figure 3, see Table 5 and 6.

Reviewer 3 Report

This manuscript presents how non-woody and woody debris affect metal elements in headwater streams in an alpine forest environment in southwestern China.  The findings of this paper are important for assessing and managing the water quality of headwater streams in this environment.  Minor revisions:

-          Section 2.1: A map showing the study site would be helpful for the reader

-          Lines 80-82:  What was the time series of these averages, min and max?

-          Lines 82-84: Can you provide few numbers about the flow regime (min, max, average)?

-          Lines 93-94: Maybe it is  better to present the characteristics of the streams in a table. Also, as a reader I would like to have more information about the slope, altitude and substrate. Especially substrate seems to play a crucial role in the results and discussion section. What is the geology of the study site?

-          Lines 95-99: What was the amount of the debris used in this study? Is it representative of the natural conditions?

-          Lines 100-117: Few pictures of the study design would be really helpful for the reader

-          Section 2.2: Maybe it is better to number the equations.

-          Lines 127-130: Why these metals were selected for further analysis? The authors in the introduction section refer to other important metals too (Na and Ca) but they are not included in the analysis.

-          Lines 152-154: I would add this table to the results section.  Also, it would be good to see the range of the water characteristics (not only the averages).

-          Section 3: I believe that it would be more convenient for the reader to reorder the figures and the tables according to the text. Maybe place each figure and table under the representative section?

-          Section 4: As a reader, I would like to see in the discussion section what do all the negative and positive correlations of the metals with the environmental variables described in the results section, mean. Also, a discussion is made about the sediments but we do not have any information about them in the materials and methods section. Are the findings of this paper applicable to all the substrates or just to areas with similar substrate?

Author Response

Dear editor

 We are very grateful for your comments and useful suggestions on our manuscript entitled "Woody litter increases headwater stream metal export in an alpine forest", (manuscript number: forests-470383). Those comments are valuable and very helpful for revising and improving our manuscript, as well as the important guiding significance to our research. Based on these comments and suggestions, we have revised the manuscript accordingly. The language has been further polished by American Journal Experts, which is an English language editing company (https://www.aje.com). All comments and suggestions provided by the reviewers are shown in black, and our responses are in blue. Revisions are marked in red in the main text. Our responses to each and all the comments are listed was follows:

 ü  Section 2.1: A map showing the study site would be helpful for the reader

Response: We thank you for the insightful comments and useful suggestions! A map showing the study site has been added in the main text as Figure 1.

ü  Lines 80-82:  What was the time series of these averages, min and max?

Response: It is the maximum and minimum.

ü  Lines 82-84: Can you provide few numbers about the flow regime (min, max, average)?

Response: Here were the climatic data from our previous experiments, and more data about flow would be monitored in further experiments.

ü  Lines 93-94: Maybe it is better to present the characteristics of the streams in a table. Also, as a reader I would like to have more information about the slope, altitude and substrate. Especially substrate seems to play a crucial role in the results and discussion section. What is the geology of the study site?

Response: Unfortunately, we did not measure the slope of streams. We have added as much information of stream characteristics as possible to the revised manuscript. All artificially excavated streams were on the same riparian zone of the original stream, with the same slope, aspect, altitude and substrate. These preconditions ensure that the results of our study are mainly affected by different litter input conditions.

ü  Lines 95-99: What was the amount of the debris used in this study? Is it representative of the natural conditions?

Response: We used a natural representative stream as a reference to add woody litter that naturally falling in the nearby riparian zone to the six artificially excavated streams, and we consider that can represent natural conditions.

ü  Lines 100-117: Few pictures of the study design would be really helpful for the reader

Response: We have added a new figure 1 as the schematic diagram of experimental design to make that easier to understand.

ü  Section 2.2: Maybe it is better to number the equations.

Response: Corrected as suggested.

ü  Lines 127-130: Why these metals were selected for further analysis? The authors in the introduction section refer to other important metals too (Na and Ca) but they are not included in the analysis.

Response: These metals were chosen because they are representative, including macroelement, microelement and heavy metal.

ü  Lines 152-154: I would add this table to the results section.  Also, it would be good to see the range of the water characteristics (not only the averages).

Response: We have modified Table 1, and added standard deviation.

ü  Section 3: I believe that it would be more convenient for the reader to reorder the figures and the tables according to the text. Maybe place each figure and table under the representative section?

Response: Corrected as suggested.

ü  Section 4: As a reader, I would like to see in the discussion section what do all the negative and positive correlations of the metals with the environmental variables described in the results section, mean. Also, a discussion is made about the sediments but we do not have any information about them in the materials and methods section. Are the findings of this paper applicable to all the substrates or just to areas with similar substrate?

Response: Our original experimental idea included that different litter input can affect the environmental factors in the stream water, and thus affecting the metal in the stream. But results found that different litter input had no significant effect on these factors, and the correlation between metals and these factors did not perform well. So, environmental factors were not the focus of our discussion. We referred to the sampling, determination, and calculation methods for surface sediments in the materials and methods section, and see revised manuscript. The influence of the substrate was not the focus of our research, and in order to eliminate the effect of the substrate, all excavated streams had the same substrate and are the mixing of clay, fine sand and gravel.

Round  2

Reviewer 1 Report

The manuscript has been improved a lot. There are, however, few points to be taken into consideration:

 The sketch presenting the experimental set-up explains a lot. However, two important elements should be added. Firstly, the points where the water samples were taken; secondly up- and downstream streams (ll. 116 – 117) should be marked because their description in the text is somewhat murky. Please see the appendix at the end of the annotated manuscript.

 The criterion dividing woody and non-woody debris (2 millimetres in diameter) does not seem well chosen. I would suspect that many flowers, fruit, and pieces of bark exceeded this limit. Were they included into the woody debris category?

 Check your units in the formulae 1 – 5. Take (1) as an example: if you express metal concentration in [µg/L] it amounts to expressing it in [mg/m3], hence your E will be in [mg/day].

 The figures look better, however not everyone will be reading your paper on-screen. Try to print/display your graphs in black-and-white and then judge how easy is it do discern the curves. Making the figures bigger (=fewer on a single page) is the simplest way to help.

 The statement that woody litter increases the flow discharge is counterintuitive. Generally, the discharge depends on the amount of water carried by a watercourse, and large stumps, branches and the like have little direct impact on that amount. The impact you mention probably occurs locally, where the configuration of the streambed in conjunction with woody litter causes acceleration of the current. Whether I am right here or not, the problem should be discussed in the paper.

Author Response

Dear editor

We are very grateful for your further comments and useful suggestions on our manuscript entitled "Woody litter increases headwater stream metal export ratio in an alpine forest", (manuscript number: forests-470383). Those comments are valuable and very helpful for revising and improving our manuscript, as well as the important guiding significance to our research. Based on these comments and suggestions, we have revised the manuscript accordingly. The language has been further polished by American Journal Experts, which is an English language editing company (https://www.aje.com). All comments and suggestions provided by the reviewers are shown in black, and our responses are in blue. Revisions are marked in red in the main text. Our responses to each and all the comments are listed was follows:

The sketch presenting the experimental set-up explains a lot. However, two important elements should be added. Firstly, the points where the water samples were taken; secondly up- and downstream streams (ll. 116 – 117) should be marked because their description in the text is somewhat murky. Please see the appendix at the end of the annotated manuscript.

Response: We thank you for the insightful comments and useful suggestions! Corrected as suggested.

The criterion dividing woody and non-woody debris (2 millimetres in diameter) does not seem well chosen. I would suspect that many flowers, fruit, and pieces of bark exceeded this limit. Were they included into the woody debris category?

Response: We consulted the relevant literature and gave a more detailed description of the classification criteria. We modified the original “bark” to “small bark fragments”. According to classification method of Harmon et. al., non-woody debris is fine litter, including fine roots, leaves, twigs, reproductive parts (including cones), and small bark fragments.

Harmon, M.E., Nadelhoffer, K.J., Blair, J.M. Measuring decomposition, nutrient turnover, and stores in plant litter. In: Robertson GP, Coleman DC, Bledsoe CS, Sollins P (eds) Standard methods for long-term ecological research. Oxford University Press, New York, pp 202-240, 1999.

Check your units in the formulae 1 – 5. Take (1) as an example: if you express metal concentration in [µg/L] it amounts to expressing it in [mg/m3], hence your E will be in [mg/day].

Response: Corrected as suggested.

The figures look better, however not everyone will be reading your paper on-screen. Try to print/display your graphs in black-and-white and then judge how easy is it do discern the curves. Making the figures bigger (=fewer on a single page) is the simplest way to help.

Response: We had separated and enlarged the figures as suggested, but we were not going to use black-and-white graphs because authors are encouraged to prepare figures and schemes in color in forests.

The statement that woody litter increases the flow discharge is counterintuitive. Generally, the discharge depends on the amount of water carried by a watercourse, and large stumps, branches and the like have little direct impact on that amount. The impact you mention probably occurs locally, where the configuration of the streambed in conjunction with woody litter causes acceleration of the current. Whether I am right here or not, the problem should be discussed in the paper.

Response: In our study, we measured the flow discharge of these streams, and the results showed that the input of woody litter significantly increased the flow discharge (Table 1). Meanwhile, we mentioned in the discussion that this may be because the input of woody litter changed the channel characteristics of streams and formed a step, which caused the increase of flow discharge.

Reviewer 2 Report

The revised manuscript of Liang et al. examines the effects of woody and non-woody debris on metal export and storage in headwater streams. I commend the authors for addressing most of my concerns listed previously. Figure 1 of the excavated channel is particularly helpful and cleared up much of my confusion regarding the experimental design. However, I still have major concerns on the statistical tests used, and interpretation of results, outlined below.

Major Concerns

My major concern stems from the interpretation of the results, particularly statements such as L 267 which states that “input of woody litter had significant effects on the metal export of water”. However, table 3, which I believe shows the statistical results of metals export, shows no significant response of litter input type. Furthermore, the right side of figure 3 shows export through time. Although the woody debris treatments (green and brown color) have a greater magnitude than the origin (blue color) the error bars are quite large, and appear to overlap for some elements tested. Therefore, it is unclear what statistical tests support the claim on L 267.

Figure 2 shows that treatments with woody debris (green and purple) have a higher export ratio than the origin stream. However, no information on statistical testing for these results is provided. As written, it appears that figure 2 is only for qualitative comparison, and therefore does not support the claim that woody debris “significantly” altered metal export.

If the magnitude of export was greater in treatment streams but not statistically different, I suggest rewording results and discussion as “Export in [treatment] streams was greater than in origin, but this difference was not significant” or similar.

It is possible that other statistical tests support these interpretations, but as written it is unclear how the work performed here supports these claims.

My other concern stems from the methods described in L 165-169. The sentences starting on lines 165 and 167 seem to be both testing the same things e.g. testing time and litter input type on concentration of water and sediment, as well as export and storage. Perhaps the sentence on L165 is actually testing for differences in water characteristics i.e. results in table 1?

The sentence on L 167 states that repeated measure 2-way ANOVA and Tukey HSD was performed. Generally speaking, an ANOVA is used to test for significant differences between a group of means. E.g. mean metal export in at least one stream type was significantly different. However, it does not give you information on which treatment mean is different. Therefore, a post-hoc Tukey HSD test is used to determine which group mean is significantly different. Generally speaking, if the ANOVA test is not significant, post-hoc tests should not be performed.

As I mentioned in my original comments, the results of post-hoc Tukey Tests are not presented in this manuscript. It is unclear what those results were, and based on the interpretation in the discussion section, I have concerns that these statistical tests were performed correctly.

These issues need to be addressed. Specifically clarification of what is being tested in L 165 and L167, and how exactly the Tukey test is being performed, and what those results are.

L 163 – I think your hypothesis needs to be more clearly defined. As written, it appears that the only hypothesis you are testing that the input of woody litter has a greater contribution to metal concentration and export than non-woody debris? Wouldn’t a more inclusive set of hypotheses be that there are differences in metal export and storage based on the type of woody and non-woody litter? If you are only testing that woody litter has greater effects than non-woody litter, why does your study design include litter exlcusion and both woody and non-woody litter?

Minor

Table 1 – are the letters here from the Tukey HSD test? Need to explain what the variation is in the caption. E.g. mean± standard deviation

Figure 2 – were any statistical tests run on these results? Linear regression? Are the lines presented significantly different from each other? E.g. slope or intercept? Or is this simply for qualitative analysis showing differences between origin stream and treatment types?

L 202 – I believe this sentence refers to figure 3, not figure 2

L 267 – the results presented here do not support the claim that woody litter significantly affects metal export in water.

L 279 – again, need to support claim of that woody litter increased export.

L 292- the right side of figure 3 shows export, but it does not appear that exclusion and only woody debris (red and yellow) are different from each other e.g. error bars overlap. Please support claim that streams with litter exclusion had higher export than streams with only woody debris in results section.

L 345 – The statement that woody litter significantly increases export is not supported based on your results.

Table 3 – I think that “output” should be “export”?

Author Response

Dear editor

We are very grateful for your further comments and useful suggestions on our manuscript entitled "Woody litter increases headwater stream metal export ratio in an alpine forest", (manuscript number: forests-470383). Those comments are valuable and very helpful for revising and improving our manuscript, as well as the important guiding significance to our research. Based on these comments and suggestions, we have revised the manuscript accordingly. The language has been further polished by American Journal Experts, which is an English language editing company (https://www.aje.com). All comments and suggestions provided by the reviewers are shown in black, and our responses are in blue. Revisions are marked in red in the main text. Our responses to each and all the comments are listed was follows:

My major concern stems from the interpretation of the results, particularly statements such as L 267 which states that “input of woody litter had significant effects on the metal export of water”. However, table 3, which I believe shows the statistical results of metals export, shows no significant response of litter input type. Furthermore, the right side of figure 3 shows export through time. Although the woody debris treatments (green and brown color) have a greater magnitude than the origin (blue color) the error bars are quite large, and appear to overlap for some elements tested. Therefore, it is unclear what statistical tests support the claim on L 267.

Response: We thank you for the insightful comments and useful suggestions! We had added the results of multiple comparisons after the tables as suggested, and modified the relevant parts in the result, discussion and conclusion. Although the effects of different litter input conditions on metal export were not significant, the input of woody litter increased the metal export ratio.

Figure 2 shows that treatments with woody debris (green and purple) have a higher export ratio than the origin stream. However, no information on statistical testing for these results is provided. As written, it appears that figure 2 is only for qualitative comparison, and therefore does not support the claim that woody debris “significantly” altered metal export.

Response: The results in Figure 2 (Figure 4, now) were tested statistically and shown in Table 5 in the revised manuscript.

If the magnitude of export was greater in treatment streams but not statistically different, I suggest rewording results and discussion as “Export in [treatment] streams was greater than in origin, but this difference was not significant” or similar.

Response: We had modified the relevant parts, which were modified as “the input of woody litter increased the metal export ratio”.

It is possible that other statistical tests support these interpretations, but as written it is unclear how the work performed here supports these claims.

Response: We had enriched the results section, please see the revised manuscript.

My other concern stems from the methods described in L 165-169. The sentences starting on lines 165 and 167 seem to be both testing the same things e.g. testing time and litter input type on concentration of water and sediment, as well as export and storage. Perhaps the sentence on L165 is actually testing for differences in water characteristics i.e. results in table 1?

Response: Corrected as suggested, please see the revised manuscript.

The sentence on L 167 states that repeated measure 2-way ANOVA and Tukey HSD was performed. Generally speaking, an ANOVA is used to test for significant differences between a group of means. E.g. mean metal export in at least one stream type was significantly different. However, it does not give you information on which treatment mean is different. Therefore, a post-hoc Tukey HSD test is used to determine which group mean is significantly different. Generally speaking, if the ANOVA test is not significant, post-hoc tests should not be performed.

As I mentioned in my original comments, the results of post-hoc Tukey Tests are not presented in this manuscript. It is unclear what those results were, and based on the interpretation in the discussion section, I have concerns that these statistical tests were performed correctly.

These issues need to be addressed. Specifically clarification of what is being tested in L 165 and L167, and how exactly the Tukey test is being performed, and what those results are.

Response: We had added the results of multiple comparisons after the tables as suggested, please see the revised manuscript.

L 163 – I think your hypothesis needs to be more clearly defined. As written, it appears that the only hypothesis you are testing that the input of woody litter has a greater contribution to metal concentration and export than non-woody debris? Wouldn’t a more inclusive set of hypotheses be that there are differences in metal export and storage based on the type of woody and non-woody litter? If you are only testing that woody litter has greater effects than non-woody litter, why does your study design include litter exlcusion and both woody and non-woody litter?

Response: Corrected as suggested.

Table 1 – are the letters here from the Tukey HSD test? Need to explain what the variation is in the caption. E.g. mean± standard deviation

Response: Corrected as suggested.

Figure 2 – were any statistical tests run on these results? Linear regression? Are the lines presented significantly different from each other? E.g. slope or intercept? Or is this simply for qualitative analysis showing differences between origin stream and treatment types?

Response: The results in Figure 2 (Figure 4, now) were tested statistically and shown in Table 5 in the revised manuscript.

L 202 – I believe this sentence refers to figure 3, not figure 2

Response: I’m sorry that there was a problem with the sequence of figures in the last revision. We had reordered the figures in new revised manuscript.

L 267 – the results presented here do not support the claim that woody litter significantly affects metal export in water.

L 279 – again, need to support claim of that woody litter increased export.

Response: We had modified the relevant parts, which were modified as “the input of woody litter increased the metal export ratio”.

L 292- the right side of figure 3 shows export, but it does not appear that exclusion and only woody debris (red and yellow) are different from each other e.g. error bars overlap. Please support claim that streams with litter exclusion had higher export than streams with only woody debris in results section.

Response: We had modified the “export” here to “export ratio”, and the description should come from Figure 4 in the revised manuscript.

L 345 – The statement that woody litter significantly increases export is not supported based on your results.

Response: We had modified the relevant parts in result, discussion and conclusion, and modified “the input of woody litter increased the metal export ratio”.

Table 3 – I think that “output” should be “export”?

Response: Corrected as suggested. Thank you again for the insightful comments and useful suggestions. We hope you will satisfy our revisions.

This manuscript is a resubmission of an earlier submission. The following is a list of the peer review reports and author responses from that submission.

Round  1

Reviewer 1 Report

This study evaluated the impact of artificial streams and artificial inputs of woody and non-woody debris in these streams and the subsequent effect on metals (K, Mg, Fe, and Cr).  The English needs substantial work throughout.  Every paragraph has at least a couple of sentences that need to be edited.  This distracts from the message and makes some of the meaning unclear. However, I focused on the science rather than the writing and will let the editors worry about the grammar.  My major concern is the study focused on one stream and that the experimental design is unclear.      

27-29. How does this research contribute to this understanding? Provide examples or explicit statements rather than a generalized overview statement.    

35.  What trace metals require attention?

44-48. Limited research is somewhat valid, but not completely valid for why it’s important to do a study.  Maybe it’s just not a big factor so people are not concerned.  

58.  Faster than what? 

90-99. This paragraph is the impetus for the entire study design and it is not clear what was done.

Are there 3 channels coming off of the natural channel? How long are they? How close are they? Each of the 4 input conditions are replicated in each artificial stream? In what order? What is the flow (volume) in these artificial streams?  Are they all the same?  Do they fluctuate greatly? How much material was put in?  How was it standardized?  What are the species?  What are the sizes?  How does this compare with natural inputs? When was all of this conducted?

What do you mean by restoration?  Why was one stream shaded? Are the streams in shaded conditions to begin with?  What was the purpose of the intercepting dams?  Was this done on all 3 streams?

Even with the lack of clarity it is clear that there is only one stream source here so the study inference is limited to this stream and not streams in general. It would seem the different treatments are cumulative.  A better approach would be one stream—one treatment. The lack of clarity in this paragraph clouds the interpretation of the rest of the manuscript and makes it difficult to know what is true or untrue.

102-104. Very short experiment (temporal timeframe).  Can’t say too much definitive about the results.  

Table 1.  Are these numbers from the water or the sediment?  

Tables 2-5.  I’m never a fan of tables with Anova’s.  Maybe better as an appendix.

Tables 6-7.  You have low r values but many significant values.  How can you count each reading as independent?  That makes little sense to me.

225. Are these levels (30-80%) anything to worry about? Or is this level helpful to aquatic productivity?

254.  Your results do not support this statement.  Results indicate no difference between woody debris and woody debris and non-woody debris input.

325.  It would be nice to provide some conclusions.  I’m left with the feeling of “so what”!  Make some sense of what you have found. 

I wish you well with your manuscript.

Author Response

Dear reviewer

 We are very grateful for your comments and useful suggestions on our manuscript entitled "Woody debris increases headwater stream metal export in an alpine forest", (manuscript number: forests-446999). Those comments are valuable and very helpful for revising and improving our manuscript, as well as the important guiding significance to our research. Based on these comments and suggestions, we have revised the manuscript thoroughly. Our responses to each and all the comments are listed below.

 ü  This study evaluated the impact of artificial streams and artificial inputs of woody and non-woody debris in these streams and the subsequent effect on metals (K, Mg, Fe, and Cr). The English needs substantial work throughout. Every paragraph has at least a couple of sentences that need to be edited. This distracts from the message and makes some of the meaning unclear. However, I focused on the science rather than the writing and will let the editors worry about the grammar. My major concern is the study focused on one stream and that the experimental design is unclear.

Response: Thank you very much for your valuable comments! We have rewritten this article and especially in more detail describing the experimental design. This article has been edited through American Journal Experts to improve the flow and readability.

ü  27-29. How does this research contribute to this understanding? Provide examples or explicit statements rather than a generalized overview statement.

Response: Thank you for the nice comments! We have re-edited the text to show our meaning more clearly.

ü  35. What trace metals require attention?

Response: Thank you for the nice comments! We have rewritten the introduction to make it more clearly, and see the revised draft.

ü  44-48. Limited research is somewhat valid, but not completely valid for why it’s important to do a study. Maybe it’s just not a big factor so people are not concerned.

Response: Thank you for the nice comments! We have rewritten the introduction, and see the revised draft.

ü  58.Faster than what?

Response: Thank you for the nice comments! We have rewritten the introduction to make it more clearly, and see the revised draft.

ü  90-99. This paragraph is the impetus for the entire study design and it is not clear what was done.

Response: Thank you for the nice comments! We have rewritten the experimental design to make it more detailed and easy to understand.

ü  Are there 3 channels coming off of the natural channel? How long are they? How close are they? Each of the 4 input conditions are replicated in each artificial stream? In what order? What is the flow (volume) in these artificial streams? Are they all the same? Do they fluctuate greatly? How much material was put in? How was it standardized? What are the species? What are the sizes? How does this compare with natural inputs? When was all of this conducted?

Response: Thank you for the nice comments! We have rewritten the experimental design, as shown in line 89-118 of the revised draft.

ü  What do you mean by restoration? Why was one stream shaded? Are the streams in shaded conditions to begin with? What was the purpose of the intercepting dams? Was this done on all 3 streams?

Response: Thank you for the nice comments! We have rewritten the experimental design, as shown in line 101-108 of the revised draft.

ü  Even with the lack of clarity it is clear that there is only one stream source here so the study inference is limited to this stream and not streams in general. It would seem the different treatments are cumulative. A better approach would be one stream—one treatment. The lack of clarity in this paragraph clouds the interpretation of the rest of the manuscript and makes it difficult to know what is true or untrue.

Response: Thank you for the nice comments! We have rewritten the experimental design to make it more detailed and clear.

ü  102-104. Very short experiment (temporal timeframe). Can’t say too much definitive about the results.

Response: Thank you for the nice comments! The purpose of this experiment is the effects of plant debris on metal elements in the forested headwater stream. According to previous research and phenological observations, the growing season in this region is from May to October. After entering the snow season, restricted flow and less fallen litter will affect plant debris inputs. It is not consistent with the experiment. Therefore, only the samples from June to October were selected in this experiment.

ü  Table 1. Are these numbers from the water or the sediment?

Response: Thank you for the nice comments! We have re-edited the Table 1. These numbers come from the water. Water is the focus of our research.

ü  Tables 2-5. I’m never a fan of tables with Anova’s. Maybe better as an appendix.

Response: Thank you for the nice comments! We have abridged Tables, and see the revised draft.

ü  Tables 6-7. You have low r values but many significant values. How can you count each reading as independent? That makes little sense to me.

Response: Thank you for the nice comments! The input of plant debris may change the characteristics of these streams. We want to see the influence of plant debris input on the streams through the correlation.

ü  225. Are these levels (30-80%) anything to worry about? Or is this level helpful to aquatic productivity?

Response: Thank you for the nice comments! We have recreated Figure 2, hoping to reflect the effect of plant debris input.

ü  254. Your results do not support this statement. Results indicate no difference between woody debris and woody debris and non-woody debris input.

Response: Thank you for the nice comments! We have rewritten the results and discussion, and see the revised draft.

ü  325. It would be nice to provide some conclusions. I’m left with the feeling of “so what”! Make some sense of what you have found.

Response: Thank you for the nice comments! We have rewritten the conclusions, and see the revised draft.

Reviewer 2 Report

The authors manipulated organic matter inputs into artificial streams and measured the accumulation and output of metals and found some significant differences among treatments. Although the research is potentially interesting to readers of Forests, the manuscript is way below standards, to the point that it is difficult to understand. Therefore, it needs to be rewritten and checked by a native English speaker before it can be properly reviewed. Even if the text is better written, I am afraid I cannot commend it for publication in Forests. I explain below some of my main concerns.

 Introduction

This section utterly fails in its goal, which is to present the state of the art of the topic and to justify objectives and hypotheses. True, metals cause problems in some streams, true, organic matter inputs are important resources, but so what? The reader is left wondering what is the purpose of the whole research. Especially, the authors should justify their hypothesis that woody debris will have a stronger effect on metal than non-woody debris, as there is nothing in the intro pointing in this direction. 

 Additionally, there are important mistakes in the intro. In lines 49-50 the authors state that metals are essential sources of energy, which is plainly wrong. Some metals (but not all) are micronutrients, so they are essential for living organisms, but none is a source of energy, unless the authors want to enter in fields such as oxidation of ferrous iron, whose metabolic importance in most streams is probably negligible. 

 Also, the authors state that non-woody debris decomposing in the stream can release large amounts of metals. Is this true? Which is the metal content of forest litter? It seems difficult to believe that it can release large amounts of metal. This sentence desperately needs to be backed up by some reference. 

 Material and methods

This section needs strong improvement. It is especially important to describe properly the experimental setting. As I think to understand (but am not sure of being right), the authors diverted water from a mountain stream to feed a series of artificial stream channels, 0.5 m width and 0.1 m deep. Please, explain how many channels were there (5 treatments x 3 replicates = 15 channels? please, confirm), how long they were, what was the water velocity, what was the substrate type, how were the channels assigned to different treatments, etc. Especially important, the authors must explain the origin and characteristics of the woody and non-woody debris they put into the channels, its distribution and the total amount of organic debris per channel, and how this amount compares to the amount in nearby streams. Also, the rationale behind the calculations is hard to follow. The authors measured water discharge and metal concentration, and calculated from there the flux of metals out from their experimental streams. But what about metals entering the streams? Should they not compute the difference between water inputs and outputs to really assess the effects of the debris they introduced?

 Lines 96-99 are puzzling. One channel (only one?) was covered with nylon mesh? Not the others? So, how can we be sure that the differences among channels occur because of the debris added, and not because of the mesh?

 Also, it is hard to understand why do the authors calculate metal output per surface area of the sample. Should not all streams be of the same surface area? Then, you can compute metal output simply multiplying concentration by water discharge, not?

 Again, to calculate metal storage, how do you know the mass of sediment in the stream? Sediment sampling is quite destructive, so how could you measure the mass and still run the experiment for 4 more months?

 On the other hand, what the authors call cumulative output rate seems rather to be an effect size of their experiment.

 Table 1 is too long. What is the purpose of showing monthly values? I would rather show average values across all the experiment. 

 Tables 2& 3 are puzzling, as the results for metal concentration and for metal output are very different. If the experimental streams were of the same size and had the same discharge (they did, I guess?), results output results should be equal to concentration results, not?

 Fig. 1. I cannot understand the letters on top of each bar. Usually we put letters to show treatments that do not differ significantly from each other, but I cannot see this is here the case. 

 Discussion

The discussion is very poor and offers no mechanistic explanation for the results. There is instead too much unsupported speculation. The authors should make it clear what is the source of the metals in their experiment. Do they come from weathering? Did they arrive with the organic debris? What is the concentration of metals in the debris they added to the streams? 

 There are too many ambiguous sentences. For instance, in L. 248, what biotic and abiotic chemical processes do they refer to?

 Again, in line 249, if woody debris had no effect on metal concentration, how could it affect metal output rates?

 L. 297-299. Authors try to explain some of their results as a consequence of aeration promoted by woody debris in the channels. But their data show oxygen concentrations did not change among treatments, so the differences must have been caused by something else.

 L. 305. There is no plankton in mountain streams, much less phytoplankton blooms. 

Author Response

Dear reviewer

 We are very grateful for your comments and useful suggestions on our manuscript entitled "Woody debris increases headwater stream metal export in an alpine forest", (manuscript number: forests-446999). Those comments are valuable and very helpful for revising and improving our manuscript, as well as the important guiding significance to our research. Based on these comments and suggestions, we have revised the manuscript thoroughly. Our responses to each and all the comments are listed below.

 ü  The authors manipulated organic matter inputs into artificial streams and measured the accumulation and output of metals and found some significant differences among treatments. Although the research is potentially interesting to readers of Forests, the manuscript is way below standards, to the point that it is difficult to understand. Therefore, it needs to be rewritten and checked by a native English speaker before it can be properly reviewed. Even if the text is better written, I am afraid I cannot commend it for publication in Forests. I explain below some of my main concerns.

Response: Thank you very much for your valuable comments! We have rewritten this article and make a comprehensive revision based on comment, hoping to meet the standard. This article has been edited through American Journal Experts to improve the flow and readability.

ü  This section utterly fails in its goal, which is to present the state of the art of the topic and to justify objectives and hypotheses. True, metals cause problems in some streams, true, organic matter inputs are important resources, but so what? The reader is left wondering what is the purpose of the whole research. Especially, the authors should justify their hypothesis that woody debris will have a stronger effect on metal than non-woody debris, as there is nothing in the intro pointing in this direction.

Response: Thank you for the nice comments! We have rewritten the introduction to make it more clearly, and see the revised draft.

ü  Additionally, there are important mistakes in the intro. In lines 49-50 the authors state that metals are essential sources of energy, which is plainly wrong. Some metals (but not all) are micronutrients, so they are essential for living organisms, but none is a source of energy, unless the authors want to enter in fields such as oxidation of ferrous iron, whose metabolic importance in most streams is probably negligible.

Response: Thank you for the nice comments! We have rewritten the introduction and modified the errors in response to your comments to ensure they are properly expressed.

ü  Also, the authors state that non-woody debris decomposing in the stream can release large amounts of metals. Is this true? Which is the metal content of forest litter? It seems difficult to believe that it can release large amounts of metal. This sentence desperately needs to be backed up by some reference.

Response: Thank you for the nice comments! We have rewritten the introduction. We are sorry that our expression is a bit exaggerated, but some studies have proved that litter contains a certain amount of metal elements in alpine forests, the litter decomposition in water is faster than that on the forest floor, and these rapidly released metal elements will have an impact on water quality.

ü  This section needs strong improvement. It is especially important to describe properly the experimental setting. As I think to understand (but am not sure of being right), the authors diverted water from a mountain stream to feed a series of artificial stream channels, 0.5 m width and 0.1 m deep. Please, explain how many channels were there (5 treatments x 3 replicates = 15 channels? please, confirm), how long they were, what was the water velocity, what was the substrate type, how were the channels assigned to different treatments, etc. Especially important, the authors must explain the origin and characteristics of the woody and non-woody debris they put into the channels, its distribution and the total amount of organic debris per channel, and how this amount compares to the amount in nearby streams. Also, the rationale behind the calculations is hard to follow. The authors measured water discharge and metal concentration, and calculated from there the flux of metals out from their experimental streams. But what about metals entering the streams? Should they not compute the difference between water inputs and outputs to really assess the effects of the debris they introduced?

Response: Thank you for the nice comments! We have rewritten the experimental design to make it more detailed and clear, and see the revised draft.

ü  Lines 96-99 are puzzling. One channel (only one?) was covered with nylon mesh? Not the others? So, how can we be sure that the differences among channels occur because of the debris added, and not because of the mesh?

Response: Thank you for the nice comments! We have rewritten the experimental design, as shown in line 101-108 of the revised draft.

ü  Also, it is hard to understand why do the authors calculate metal output per surface area of the sample. Should not all streams be of the same surface area? Then, you can compute metal output simply multiplying concentration by water discharge, not?

Response: Thank you for the nice comments! Corrected as suggested.

ü  Again, to calculate metal storage, how do you know the mass of sediment in the stream? Sediment sampling is quite destructive, so how could you measure the mass and still run the experiment for 4 more months?

Response: Thank you for the nice comments! The surface sediment was calculated in this experiment. We regarded the surface sediment as a cuboid, measured its length, width and depth in the stream, and took a fixed volume of surface sediments back to determine its water content, dry mass, and then calculated its density and quality. It is true that the sampling of sediments is destructive, but the headwater stream has a high sediment yields. We minimized the destruct to surface sediments during the sampling process, and the sampling amount of sediments in each experimental stream was the same to ensure overall consistency.

ü  On the other hand, what the authors call cumulative output rate seems rather to be an effect size of their experiment.

Response: Thank you for the nice comments! We have revised this part, and see the revised draft.

ü  Table 1 is too long. What is the purpose of showing monthly values? I would rather show average values across all the experiment.

Response: Thank you for the nice comments! Corrected as suggested.

ü  Tables 2& 3 are puzzling, as the results for metal concentration and for metal output are very different. If the experimental streams were of the same size and had the same discharge (they did, I guess?), results output results should be equal to concentration results, not?

Response: Thank you for the nice comments! Although the metal export in the water was calculated from the concentration, the export was different from the concentration because the input of plant debris changed the flow discharge.

ü  Fig. 1. I cannot understand the letters on top of each bar. Usually we put letters to show treatments that do not differ significantly from each other, but I cannot see this is here the case.

Response: Thank you for the nice comments! We have modified Figure 1, and see the revised draft.

ü  The discussion is very poor and offers no mechanistic explanation for the results. There is instead too much unsupported speculation. The authors should make it clear what is the source of the metals in their experiment. Do they come from weathering? Did they arrive with the organic debris? What is the concentration of metals in the debris they added to the streams?

Response: Thank you for the nice comments! We have rewritten the discussion, and this section focused on the effects of plant debris input on metal elements in the water and the sediment in the headwater stream.

ü  There are too many ambiguous sentences. For instance, in L. 248, what biotic and abiotic chemical processes do they refer to?

Response: Thank you for the nice comments! Corrected as suggested.

ü  Again, in line 249, if woody debris had no effect on metal concentration, how could it affect metal output rates?

Response: Thank you for the nice comments! The metal element export in the water depended on the metal element concentration in the water and the flow discharge. The input of woody debris increased the flow discharge and resulted in more metal element export.

ü  L. 297-299. Authors try to explain some of their results as a consequence of aeration promoted by woody debris in the channels. But their data show oxygen concentrations did not change among treatments, so the differences must have been caused by something else.

Response: Thank you for the nice comments! We have modified Table 1 and shown average values across all the experiment. In this way it clearly showed the effect of the input of plant debris on dissolved oxygen in the headwater stream.

ü  L. 305. There is no plankton in mountain streams, much less phytoplankton blooms.

Response: Thank you for the nice comments! We have rewritten the discussion, and see the revised draft.